# Linear Convergence of Sinkhorn's Algorithm for Generalized Static Schrödinger Bridge

**Rahul Choudhary** [1]  **Hanbaek Lyu** [1]

## Abstract

The classical static Schrödinger Bridge (SSB) problem, which seeks the most likely stochastic evolution between two marginal probability measures, has been studied extensively in the optimal transport and statistical physics communities, and more recently in machine learning communities in the surge of generative models. The standard approach to solve SSB is to first identify its Kantorovich dual and use Sinkhorn's algorithm to find the optimal potential functions. While the original SSB is only a strictly convex minimization problem, this approach is known to warrant linear convergence under mild assumptions. In this work, we consider a generalized SSB allowing any strictly increasing divergence functional, far generalizing the entropy functional $x \log x$ in the standard SSB. This problem naturally arises in a wide range of seemingly unrelated problems in entropic optimal transport, random graphs/matrices, and combinatorics. We establish Kantorovich duality and linear convergence of Sinkhorn's algorithm for the generalized SSB problem under mild conditions. Our results provide a new rigorous foundation for understanding Sinkhorn-type iterative methods in the context of large-scale generalized Schrödinger bridges.

## 1. Introduction

Fix an $m \times n$ real nonnegative matrix $\mathbf{W}$. A pair of vectors $(\mathbf{r}, \mathbf{c}) \in \mathbb{R}^m \times \mathbb{R}^n$ is called a *margin* if we have the same total sum: $\langle \mathbf{r}, \mathbf{1} \rangle = \langle \mathbf{c}, \mathbf{1} \rangle$. Fix a twice-differentiable strictly convex function $f : \mathbb{R} \to \mathbb{R} \cup \{\infty\}$[1]. Our central interest in this

work is to solve the following matrix optimization problem:

$$\mathbf{Z}^{\mathbf{r},\mathbf{c}} := \operatorname*{argmin}_{\mathbf{X}=(x_{ij}) \in \mathbb{R}^{m \times n}} \left[ g(\mathbf{X}) := \sum_{i,j} f(x_{ij}) \mathbf{W}_{ij} \right] \quad (1)$$

$$\text{subject to } \sum_{j=1}^{n} x_{ij} \mathbf{W}_{ij} = \mathbf{r}(i), \ \sum_{i=1}^{m} x_{ij} \mathbf{W}_{ij} = \mathbf{c}(j) \ \forall i, j.$$

We call the optimal matrix $\mathbf{Z}^{\mathbf{rc}}$ above the *(generalized) static Schrödinger bridge (SSB)* for *margin* $(\mathbf{r}, \mathbf{c})$ w.r.t. weight $\mathbf{W}$ and *divergence* $f$. The entries of $\mathbf{Z}^{\mathbf{r},\mathbf{c}}$ are taken to be arbitrary where $w_{ij} = 0$. We call the set $\mathcal{T}_{\mathbf{W}}(\mathbf{r}, \mathbf{c})$ of all real $m \times n$ matrices $\mathbf{X} = (x_{ij})$ satisfying the margin constraint in (1) the *(**W**-weighted) transportation polytope* with margin $(\mathbf{r}, \mathbf{c})$. The matrix optimization problem above enjoys a rich connection to (discrete instances of) a wide range of important problems.

First, our SSB problem (1) is exactly the *f-divergence SSB* problem formulated in terms of the relative density (Carlier et al., 2017; Lorenz & Mahler, 2022; Terjék & González-Sánchez, 2022). Suppose $\mathbf{r}$ and $\mathbf{c}$ give probability mass functions on $[m] := \{1, \ldots, m\}$ and $[n]$ and $\mathbf{W}$ gives a joint probability mass function on the lattice $[m] \times [n]$. Then $f$-divergence SSB between the margins $\mathbf{r}$ and $\mathbf{c}$ w.r.t. $\mathcal{W}$ is the probability measure $\mathcal{H}$ solving the following constrained minimization problem

$$\min_{\mathcal{H} \in \Pi(\mathbf{r}, \mathbf{c})} \left[ D_f(\mathcal{H} \| \mathcal{W}) := \int f\left( \frac{d\mathcal{H}}{d\mathcal{W}} \right) d\mathcal{W} \right], \quad (2)$$

where the minimum is for all probability measures $\mathcal{H}$ on $[m] \times [n]$ that is absolutely continuous w.r.t. $\mathcal{W}$ and has marginal density $(\mathbf{r}, \mathbf{c})$. Rewriting the above in terms of the relative density $X = \frac{d\mathcal{H}}{d\mathcal{W}} \in \mathbb{R}^{m \times n}$ gives exactly our SSB problem (1). Specializing to the KL-divergence by taking $f(x) = x \log x$, it further becomes the classical SSB between marginal densities $\mathbf{r}$ and $\mathbf{c}$ w.r.t. the reference measure $\mathcal{W}$ (see, e.g., (Fortet, 1940; Pavon et al., 2021)).

Second, if we take the KL-divergence and $\mathbf{W}$ in the following form

$$\mathbf{W}_{ij} \propto e^{-c_{ij}/\varepsilon} \mathbf{r}(i) \mathbf{c}(j) \quad (3)$$

---

*Equal contribution. [1]University of Wisconsin-Madison, Madison, Wisconsin. Correspondence to: Rahul Choudhary <rahul.choudhary@wisc.edu>, Hanbaek Lyu <hlyu@math.wisc.edu>.

*Proceedings of the 42nd International Conference on Machine Learning*, Vancouver, Canada. PMLR 267, 2025. Copyright 2025 by the author(s).

[1]By Jensen's inequality, the domain of $f$, $\operatorname{dom}(f) := \{x \in \mathbb{R} | |f(x)| < \infty\}$ is an interval. We understand differentiability of

$f$ for $f$ restricted on the interior of its domain.

for some *cost matrix* $C = (c_{ij}) \in \mathbb{R}_{\geq 0}^{m \times n}$ and regularization parameter $\varepsilon > 0$, then (1) becomes the *entropic optimal transport* (EOT) problem with marginal densities $(\mathbf{r}, \mathbf{c})$ and cost matrix $C$ (Villani, 2021).

Third, if we take the KL-divergence and uniform marginals are $\mathbf{r} = m^{-1}\mathbf{1}_m$ and $\mathbf{c} = n^{-1}\mathbf{1}_n$, then (1) becomes the classical *matrix scaling* problem, which is to find the doubly stochastic matrix that minimizes the KL-divergence from $\mathbf{W}$ (Sinkhorn, 1964). Indeed, such a doubly stochastic matrix is given by $\mathbf{Z}^{\mathbf{r},\mathbf{c}} \odot \mathbf{W}$, where $\mathbf{Z}^{\mathbf{r},\mathbf{c}}$ is the SSB in (1) for the uniform margin and $\odot$ denotes the entrywise product.

Fourth, (1) is also connected to the *typical table* problem in the combinatorics literature (Barvinok, 2010), which is central to the enumeration of integer-valued nonnegative matrices with prescribed row sum $\mathbf{r}$ column sum $\mathbf{c}$ (contingency tables with margin $(\mathbf{r}, \mathbf{c})$). For this, we may take $\mathbf{W} = \mathbf{1}_m \mathbf{1}_n^\top$ and $f(x) = -(x+1)\log(x+1) + x\log x$, which is the negative of the entropy of the geometric distribution with mean $x$. The solution of the corresponding instance of (1) is called the typical table for margin $(\mathbf{r}, \mathbf{c})$ and it is known that a uniformly random contingency table with margin $(\mathbf{r}, \mathbf{c})$ is sharply concentrated around the typical table (Barvinok, 2010; Lyu & Mukherjee, 2024).

Fifth, if we take $\mathbf{r} = \mathbf{c} = \mathbf{d}$, a sequence of degrees of an $n$-node simple graph and $f(x) = x\log x + (1-x)\log(1-x)$ (negative entropy of the Bernoulli distribution with mean $x$) with $\mathbf{W} = \mathbf{1}_m \mathbf{1}_n^\top$, then the adjacency matrix of a uniformly random $n$-node graph with degree sequence $\mathbf{d}$ is concentrated to the solution of (1) in the cut metric (Chatterjee et al., 2011).

Lastly, (1) is also connected to a problem in random matrix theory. Namely, the $m \times n$ random matrix with i.i.d. entry with entrywise distribution $\mu$ conditioned to have row sum $\mathbf{r}$ and column sum $\mathbf{c}$ is known to be concentrated around the solution of (1) (a.k.a. generalized typical table) in cutnorm, in which we take $\mathbf{W} = \mathbf{1}_m \mathbf{1}_n^\top$ and $f(x)$ is the KL-divergence from $\mu$ to the exponential tilting of $\mu$ to have mean $x$ (Lyu & Mukherjee, 2024).

The overarching goal in this work is to propose a Sinkhorn-style algorithm that solves a 'Kantorovich dual' of (1). For a gentle introduction, consider writing down the multivariate Lagrange multiplier equations for (1). There are $m + n$ linear equations (in fact one of them is redundant due to the consistency condition $\langle \mathbf{r}, \mathbf{1} \rangle = \langle \mathbf{c}, \mathbf{1} \rangle$) for the (weighted) row sum ($m$) and column sum ($n$) constraints. Since the objective of (1) is strictly convex, existence of a solution implies that the solution $\mathbf{Z}^{\mathbf{r},\mathbf{c}} = (z_{ij})$ is uniquely determined by

$$f'(z_{ij}) = \boldsymbol{\alpha}(i) + \boldsymbol{\beta}(j) \qquad \text{whenever } w_{ij} \neq 0, \qquad (4)$$

where $(\boldsymbol{\alpha}, \boldsymbol{\beta}) \in \mathbb{R}^m \times \mathbb{R}^n$ are vectors of Lagrange multipliers. We call $\boldsymbol{\alpha}$ and $\boldsymbol{\beta}$ the *row potential* and *column potential*,

respectively. Since the SSB $\mathbf{Z}^{\mathbf{r},\mathbf{c}}$ is assumed to satisfy the margin constraint $\mathbf{Z}^{\mathbf{r},\mathbf{c}} \in \mathcal{T}_{\mathbf{W}}(\mathbf{r}, \mathbf{c})$, necessarily the potentials need to solve the *(generalized) Schrödinger system*:

$$\begin{cases} \sum_{j=1}^n \mathbf{W}_{ij} \cdot (f')^{-1}(\boldsymbol{\alpha}(i) + \boldsymbol{\beta}(j)) = \mathbf{r}(i) & \forall 1 \leq i \leq m \\ \sum_{i=1}^m \mathbf{W}_{ij} \cdot (f')^{-1}(\boldsymbol{\alpha}(i) + \boldsymbol{\beta}(j)) = \mathbf{c}(j) & \forall 1 \leq j \leq n. \end{cases} \quad (5)$$

Note that since $f$ is twice-differentiable and strictly convex, $f'' > 0$ so $f'$ is strictly increasing, so it admits a strictly increasing inverse $(f')^{-1}$.

In order to solve (5), we propose the following Sinkhorn-type iterative algorithm where for given potentials $(\boldsymbol{\alpha}_{k-1}, \boldsymbol{\beta}_{k-1})$ at iteration $k$, one first finds the new column potential $\boldsymbol{\alpha}_k$ that satisfies the Schrödinger system for the column sums using the current row potential $\boldsymbol{\alpha}_{k-1}$, and then updates the row potential using the updated column potential $\boldsymbol{\beta}_k$ similarly:

$$\mathbf{GSA:} \begin{cases} \forall 1 \leq j \leq n, \boldsymbol{\beta}_k(j) \leftarrow \text{unique } \beta \in \mathbb{R} \text{ s.t.} \\ \quad \mathbf{c}(j) = \sum_{i=1}^m (f')^{-1}(\boldsymbol{\alpha}_{k-1}(i) + \beta)\mathbf{W}_{ij}, \\ \forall 1 \leq i \leq m, \boldsymbol{\alpha}_k(i) \leftarrow \text{unique } \alpha \in \mathbb{R} \text{ s.t.} \\ \quad \mathbf{r}(i) = \sum_{j=1}^n (f')^{-1}(\alpha + \boldsymbol{\beta}_k(j))\mathbf{W}_{ij}. \end{cases} \quad (6)$$

Since $(f')^{-1}$ is strictly increasing, the solution to (6) for each iteration $k$ is unique if it exists.

We call the above the *generalized Sinkhorn algorithm* (GSA). Indeed, notice that when $f(x) = x\log x$ so that our main problem (1) reduces to the classical static Schrödinger bridge and entropic optimal transport, $f'(x) = 1 + \log x$ so $(f')^{-1}(\theta) = \exp(\theta - 1)$. In this case, (6) reduces to the celebrated Sinkhorn's algorithm (SA) (a.k.a. iterative proportional fitting procedure) used to solve various classical problems such as matrix scaling, static Schrödinger bridge, and entropic optimal transport (Franklin & Lorenz, 1989; Cuturi, 2013):

$$\mathbf{SA:} \begin{cases} \forall 1 \leq i \leq n, \boldsymbol{\beta}_k(j) \leftarrow \log \frac{\mathbf{c}(j)}{\sum_{i=1}^m \mathbf{W}_{ij}\exp(\boldsymbol{\alpha}_{k-1}(i))}, \\ \forall 1 \leq i \leq m, \boldsymbol{\alpha}_k(i) \leftarrow \log \frac{\mathbf{r}(i)}{\sum_{j=1}^n \mathbf{W}_{ij}\exp(\boldsymbol{\beta}_k(j))}. \end{cases} \quad (7)$$

Convergence of SA (7) has been studied extensively in the literature. Franklin and Lorenz showed that the convergence rate is exponential (linear in the log scale) in the space of margins endowed with Hilbert's projective metric (Franklin & Lorenz, 1989). A similar result was obtained for the continuous setting by Chen and Pavon (Chen et al., 2016a). Rüschendorf (Ruschendorf, 1995) established asymptotic convergence of Sinkhorn algorithm in the continuous case from the perspective of information projection. Marino and Gerolin extended a similar result to the multi-marginal case (Marino & Gerolin, 2020). Carlier established linear convergence of multi-marginal Sinkhorn algorithm in the Euclidean metric (Carlier, 2022).

All of the convergence results mentioned above concern the KL divergence ($f(x) = x \log x$) with there being no known convergence results for other divergence measures and weight matrices. Below in Theorem 2.6, we establish that the generalized Sinkhorn algorithm (6) for arbitrary divergence and weight matrix converges at a linear rate in the sense that the objective value gap decays exponentially fast.

## 1.1. Contributions

- We show that any potential functions $(\boldsymbol{\alpha}, \boldsymbol{\beta})$ characterizing the generalized static Schrödinger bridge $\mathbf{Z}^{\mathbf{r},\mathbf{c}}$ solve a dual problem, which is the Kantorovich dual to (1). See Theorem 2.4 for details.

- Under mild conditions on the divergence $f$, weight $\mathbf{W}$, and margin $(\mathbf{r}, \mathbf{c})$, we establish that the generalized Sinkhorn algorithm (6) decreases the dual objective at a linear rate. This rate may or may not be dimension independent depending on how fast the margins grow with the dimension. Theorem 2.6 covers the linear convergence result in detail.

- In order to arrive at dimension independent linear convergence, we show sufficient conditions for the boundedness of the iterates of our generalized Sinkhorn algorithm. These sufficient conditions ensure that the tameness parameter (see Def. 2.5) for the given problem margins are dimension independent. Consequently, dimension independent linear convergence of our generalized Sinkhorn algorithm ensues, making our algorithm particularly attractive for high dimensional problems. This result assumes uniformly bounded cost function. For more details, see Theorems 2.7 and 2.8.

Recently, Lyu and Mukherjee (Lyu & Mukherjee, 2024) established linear convergence of the generalized Sinkhorn algorithm (6) for the generalized typical table problem in a random matrix theory context. This is a special case of (1) where $\mathbf{W} = \mathbf{1}_m \mathbf{1}_n^\top$ and $f(x)$ is the KL-divergence from $\mu$ to the exponential tilting of $\mu$ to have mean $x$ (see Appendix A for details). Our analysis in this work is inspired by their approach and we generalize it to the abstract setting (1).

## 1.2. Related works

### 1.2.1. Schrödinger Bridge in Machine Learning

The Schrödinger Bridge (SB) problem has gained increasing attention in the machine learning community due to its connections with probabilistic modeling and optimal transport. Starting from the seminal work of Fortet (Fortet, 1940) and more recent expositions by Léonard (Léonard, 2013), the SB problem has been reinterpreted as an entropic variant of optimal transport, often called entropic optimal transport, and as such it has found applications in large-scale generative modeling.

In recent years, various approaches have been proposed to leverage Schrödinger bridges for generative modeling and density estimation. For instance, it has been shown that SB can be viewed as a stochastic control or diffusion-based process that interpolates between given marginals, often leading to dynamic formulations in the continuous-time setting (Chen et al., 2016b). More recently, several works have explored diffusion–Schrödinger bridge approaches, bringing together ideas from score-based diffusion models and the classical Schrödinger bridge problem to design generative algorithms that are both statistically efficient and robust to numerical issues arising in unregularized cost minimization (Wang et al., 2021; De Bortoli et al., 2021; Jiao et al., 2024). We refer to these works for detailed expositions on how iterative Schrödinger bridge solvers can be coupled with score-matching ideas in generative modeling.

Beyond direct use in generative methods, Schrödinger bridges also appear in other machine learning tasks involving distribution alignment or adaptation. For example, SBs can be used to perform domain adaptation where one aims to transport empirical distributions across different domains (Benamou et al., 2015), or to regularize large-scale graph alignment problems. The major appeal in all of these settings lies in the entropic regularization that smooths the transport plan, often translating into more stable optimization problems and efficient numerical methods.

### 1.2.2. Sinkhorn algorithm for static Schrödinger Bridge

At the heart of many Schrödinger bridge (and entropic optimal transport) algorithms is the Sinkhorn algorithm, also called the iterative proportional fitting procedure (IPFP). In the classical setting - where the problem is the convergence to a doubly stochastic limit of a sequence of matrices obtained from a non-negative matrix - the Sinkhorn algorithm dates back to the work of Sinkhorn and Knopp (Sinkhorn & Knopp, 1967), and has been extensively analyzed in the matrix analysis and statistics literature (Deming & Stephan, 1940; Franklin & Lorenz, 1989). Franklin and Lorenz (Franklin & Lorenz, 1989) established linear convergence (i.e., exponential decay of the error) in the Hilbert projective metric, and Rüschendorf (Ruschendorf, 1995) extended the analysis to continuous measures via an information projection viewpoint. Subsequent works by Chen and Pavon (Chen et al., 2016a) further analyzed the continuous entropic Schrödinger bridge problem with a similar iterative strategy; Marino and Gerolin (Marino & Gerolin, 2020) and Carlier (Carlier, 2022) extended convergence results to multi-marginal Sinkhorn algorithms, again establishing linear rates under reasonable conditions on the reference measures.

More recently, the Sinkhorn algorithm has become a mainstay in computational optimal transport, thanks largely to the seminal paper by Cuturi (Cuturi, 2013) that popularized the entropic regularization perspective. This line of work has led to an extensive suite of efficient implementations and theoretical studies on the stability, complexity, and generalization of Sinkhorn-based methods in high dimensions (Peyré et al., 2019). Despite these advancements, most classical convergence guarantees have focused on KL divergence (or equivalently $f(x) = x \log x$). In practice, though, there are many settings where one may wish to replace the standard KL divergence by other strictly convex divergences - leading to the generalized static Schrödinger bridge problem considered in this work.

Overall, by bridging the gap between classical matrix-scaling arguments and modern-day generative modeling, our work provides a conceptual and theoretical foundation for generalized Sinkhorn algorithms across a broad range of entropic transports and divergences.

## 2. Statements of results

Since the divergence $f : \mathbb{R} \to \mathbb{R} \cup \{\infty\}$ is convex, By Jensen's inequality, the domain of $f$, $\mathrm{dom}(f) := \{x \in \mathbb{R} \mid |f(x)| < \infty\}$ is an interval. We will write its interior as $(A, B)$ for some extended reals $A, B$. Note that $(f'(A), f'(B)) = \mathrm{Interior}(\mathrm{dom}(\psi))$.

**Assumption 2.1.** *$f$ is twice-differentiable on $(A, B)$ with $f'' > 0$ and takes $\infty$ on $\mathbb{R} \setminus [A, B]$. Furthermore, $\lim_{x \searrow A} f'(x) = -\infty$ if $|A| < \infty$; and $\lim_{x \nearrow B} f'(x) = \infty$ if $|B| < \infty$.*

Without loss of generality we can assume every row sum of $\mathbf{W}$ is positive since if, for instance, the first row of $\mathbf{W}$ is entirely zero, then we may as well omit the first row of $\mathbf{Z^{r,c}}$ entirely from our problem setting (1) and consider only the $(m-1) \times n$ submatrix. Similarly, we may assume that every column sum of $\mathbf{W}$ is also positive. Furthermore, we will impose that the weight matrix $\mathbf{W}$ satisfies the following structural assumption.

**Assumption 2.2.** *$\mathbf{W} = \mathbf{C} \odot \mathbf{u v}^\top$ for some positive $m \times n$ matrix $\mathbf{C}$ and positive vectors $\mathbf{u} \in \mathbb{R}^m$, $\mathbf{v} \in \mathbb{R}^n$, where $\odot$ denotes the Hadamard product.*

Comparing with the standard choice (3) in entropic optimal tranport, $\mathbf{C}$ plays the role of the (exponentiated) cost matrix and $\mathbf{u v}^\top$ is the prior product distribution, which is often taken as $\mathbf{r c}^\top$ from the target margin $(\mathbf{r}, \mathbf{c})$. In our analysis, the following 'condition number' of $\mathbf{C}$ will play an important role:

$$\kappa := \mathbf{C}_{\max} / \mathbf{C}_{\min} \tag{8}$$

where $\mathbf{C}_{\max}$ and $\mathbf{C}_{\min}$ are maximum and minimum entries of $\mathbf{C}$ respectively.

### 2.1. Generalized Kantorovich duality

Notice that since an SSB $\mathbf{Z^{r,c}}$ in (1), if exists, is a member of $\mathcal{T}_{\mathbf{W}}(\mathbf{r}, \mathbf{c}) \cap (A, B)^{m \times n}$. Hence this set being non-empty is a necessary condition for the existence of SSB. In the following lemma, we show that this condition is also sufficient for the existence of SSB, in which case it is also unique.

**Lemma 2.3** (Existence and uniqueness of generalized SSB). *Assume 2.1 holds. For an $m \times n$ margin $(\mathbf{r}, \mathbf{c})$ and weight matrix $\mathbf{W}$, the SSB $\mathbf{Z^{r,c}}$ in (1) exists if and only if the set $\mathcal{T}_{\mathbf{W}}(\mathbf{r}, \mathbf{c}) \cap (A, B)^{m \times n}$ is non-empty. Furthermore, the SSB is unique if it exists.*

Inspired by the Kantorovich dual for the entropic optimal transport (Nutz & Wiesel, 2022) and also the maximum likelihood formulation of the typical table problem in (Lyu & Mukherjee, 2024), we seek to construct a dual problem for (1) that is directly solved by the potentials solving the Schrödinger system (5). Let $\psi(y) = \sup_x (yx - f(x))$ denote the convex conjugate of $f$. Since $f$ is differentiable and strictly convex, the supremum for $\psi(y)$ is attained at $y = f'(x)$. Denoting $\phi := f'$, this gives

$$f(x) = x\phi(x) - \psi(\phi(x)). \tag{9}$$

Also, using $\psi' = f'' > 0$, we deduce $\psi' = \phi^{-1}$ by differentiating both sides of the above relation.

In (Lyu & Mukherjee, 2024), it was found that if $f(x) = D_{KL}(\mu_{\phi(x)} \| \mu)$, the KL-divergence from a probability measure $\mu$ and its exponential tilt $\mu_{\phi(x)}$ with mean $x$, the function $\phi$ maps tilt parameters to the mean after tilting, $\psi' = \phi^{-1}$ maps the target mean to the corresponding tilt parameters, and the convex conjugate $\psi$ of $f$ is the log-Laplace transform of $\mu$.

Now consider the following (not strictly) concave maximization problem:

$$\sup_{\boldsymbol{\alpha}, \boldsymbol{\beta}} \Big( g^{\mathbf{r,c}}(\boldsymbol{\alpha}, \boldsymbol{\beta}) := \langle \mathbf{r}, \boldsymbol{\alpha} \rangle + \langle \mathbf{c}, \boldsymbol{\beta} \rangle - \langle \mathbf{W}, \psi(\boldsymbol{\alpha} \oplus \boldsymbol{\beta}) \rangle \Big), \tag{10}$$

where $\boldsymbol{\alpha} \oplus \boldsymbol{\beta}$ denotes the $m \times n$ matrix with entries $\boldsymbol{\alpha}(i) + \boldsymbol{\beta}(j)$ and the supremum in (10) is over all $(\boldsymbol{\alpha}, \boldsymbol{\beta}) \in \mathbb{R}^m \times \mathbb{R}^n$ such that $\psi(\boldsymbol{\alpha} \oplus \boldsymbol{\beta})$ is well-defined and the inner product between matrices is the sum of the entrywise products, as usual. We will call the above the *Kantorovich dual* for (1) due to the following result. Note that its first-order optimality condition is exactly the Schrödinger system (5).

**Theorem 2.4** (Generalized Kantorovich duality). *Assume 2.1 holds. Fix an $m \times n$ margin $(\mathbf{r}, \mathbf{c})$ and weight matrix $\mathbf{W}$.*

**(i)** *Suppose the SSB $\mathbf{Z^{r,c}}$ for margin $(\mathbf{r}, \mathbf{c})$ w.r.t. $\mathbf{W}$ exists. Then there is a potential $(\boldsymbol{\alpha}, \boldsymbol{\beta})$ for $(\mathbf{r}, \mathbf{c})$ such that*

$$\mathbf{Z}_{ij} = \psi'(\boldsymbol{\alpha}(i) + \boldsymbol{\beta}(j)) \quad \forall 1 \le i \le m, 1 \le j \le n. \tag{11}$$

*Also, $(\boldsymbol{\alpha}, \boldsymbol{\beta})$ is a solution to the dual problem (10).*

**(ii)** *Suppose* $(\boldsymbol{\alpha}, \boldsymbol{\beta})$ *is a solution to the dual problem* (10). *Then the SSB* $\mathbf{Z}^{\mathbf{r},\mathbf{c}}$ *for margin* $(\mathbf{r},\mathbf{c})$ *uniquely exists and*

$$\mathbf{Z}^{\mathbf{r},\mathbf{c}} = \psi'(\boldsymbol{\alpha} \oplus \boldsymbol{\beta}). \tag{12}$$

*Furthermore, the SSB* (1) *and the dual* (10) *problems are in strong duality:*

$$\inf_{\mathbf{X} \in \mathcal{T}_{\mathbf{W}}(\mathbf{r},\mathbf{c})} g(\mathbf{X}) = \sup_{\boldsymbol{\alpha}, \boldsymbol{\beta}} g^{\mathbf{r},\mathbf{c}}(\boldsymbol{\alpha}, \boldsymbol{\beta}). \tag{13}$$

Note that the potentials for margin $(\mathbf{r},\mathbf{c})$ (defined below (4)) are not unique. In fact, if $(\boldsymbol{\alpha}, \boldsymbol{\beta})$ is a potential, then $(\boldsymbol{\alpha} + a\mathbf{1}_m, \boldsymbol{\beta} - a\mathbf{1}_n)$ is a potential for any scalar $a \in \mathbb{R}$. But it can be made unique once a further linear constraint, say $\langle \boldsymbol{\alpha}, \mathbf{1}_m \rangle = 0$, is imposed. We call such potential where the coordinates of the row potential $\boldsymbol{\alpha}$ sum to zero a *standard potential* for $(\mathbf{r},\mathbf{c})$.

## 2.2. Convergence of generalized Sinkhorn algorithm

As we noted before, the Kantorovich dual (10) is a concave maximization problem and the objective function is not strictly concave. While linear convergence of standard optimization algorithms for such problems is not usually expected, our main result (Thm. 2.6) below establishes linear convergence where the rate of convergence depends sensitively on the 'quality of the optimizer'. Namely, it will be crucial to assess how far the entries of the SSB $\mathbf{Z}^{\mathbf{r},\mathbf{c}}$ are bounded away from the boundary values $A$ and $B$, or equivalently, how far the entries of the 'direct-sum potential' $\boldsymbol{\alpha} \oplus \boldsymbol{\beta}$ are bounded away from the boundary values $\phi(A)$ and $\phi(B)$. We make this precise via the following notion of '$\delta$-tameness' of margin $(\mathbf{r},\mathbf{c})$.

**Definition 2.5** (Tame margins). Fix $\delta > 0$. We call a margin $(\mathbf{r},\mathbf{c})$ $\delta$-*tame* if the SSB w.r.t. the weight matrix $\mathbf{W}$, say $\mathbf{Z}^{\mathbf{r},\mathbf{c}}$, exists and its entries satisfy

$$A_\delta := \max\left(A + \delta, -\frac{1}{\delta}\right) \leq \mathbf{Z}^{\mathbf{r},\mathbf{c}} \leq \min\left(B - \delta, \frac{1}{\delta}\right) =: B_\delta, \tag{14}$$

or, equivalently (due to Theorem 2.4),

$$\phi(A_\delta) \leq \boldsymbol{\alpha} \oplus \boldsymbol{\beta} \leq \phi(B_\delta). \tag{15}$$

We remark that since $\mathbf{Z}^{\mathbf{r},\mathbf{c}}$ takes entries from $(A, B)$, any margin $(\mathbf{r},\mathbf{c})$ for which SSB $\mathbf{Z}^{\mathbf{r},\mathbf{c}}$ w.r.t. $\mathbf{W}$ exists, it is $\delta$-tame for some $\delta > 0$ that may depend on $(\mathbf{r},\mathbf{c})$, and hence on the dimensions $m$ and $n$.

Now we state the main result in this work. We establish that the generalized Sinkhorn algorithm (6) converges at a linear rate in the sense that the objective value gap as well as the $L^2$-estimation error of the direct-sum potentials decay exponentially fast. For the statement below, define $\|\mathbf{A}\|_{\mathbf{u}\mathbf{v}^\top} := \sqrt{\sum_{i,j} \mathbf{A}_{ij}^2 \mathbf{u}(i)\mathbf{v}(j)}$ for $m \times n$ matrix $\mathbf{A}$.

**Theorem 2.6** (Linear convergence of generalized Sinkhorn iterates). *Assume* 2.1 *and* 2.2 *hold. Let* $(\boldsymbol{\alpha}_k, \boldsymbol{\beta}_k)$, $k \geq 0$ *denote the iterates produced by the Sinkhorn algorithm* (6) *for some* $m \times n$ *margin* $(\mathbf{r},\mathbf{c})$ *such that* $\mathcal{T}_{\mathbf{W}}(\mathbf{r},\mathbf{c}) \cap (A, B)^{m \times n}$ *is non-empty. Choose* $\delta > 0$ *small enough so that* $(\mathbf{r},\mathbf{c})$ *is* $\delta$-*tame. Fix a potential* $(\boldsymbol{\alpha}^*, \boldsymbol{\beta}^*)$ *and denote* $\Delta_k := g^{\mathbf{r},\mathbf{c}}(\boldsymbol{\alpha}^*, \boldsymbol{\beta}^*) - g^{\mathbf{r},\mathbf{c}}(\boldsymbol{\alpha}_k, \boldsymbol{\beta}_k)$. *For each* $\varepsilon > 0$, *let* $\sigma_-(\varepsilon)^2$ *(resp.,* $\sigma_+(\varepsilon)^2$*) denote the infimum (resp., supremum) of* $\psi''$ *on* $(\phi(A_\varepsilon), \phi(B_\varepsilon))$. *Let* $\kappa$ *be as in* (8).

**(i)** *(Asymptotic linear convergence) There exists an integer* $k_0 \geq 0$ *that may depend on* $m, n, f, \mathbf{W}$ *such that the following holds with* $\varepsilon = \delta/2$:

$$\frac{\mathbf{C}_{\min}\sigma_-(\varepsilon)^2}{2}\|(\boldsymbol{\alpha}^* \oplus \boldsymbol{\beta}^*) - (\boldsymbol{\alpha}_k \oplus \boldsymbol{\beta}_k)\|_{\mathbf{u}\mathbf{v}^\top}^2 \leq \Delta_k \tag{16}$$

$$\leq \left(1 - \kappa^{-2}\frac{\sigma_-(\varepsilon)^4}{\sigma_+(\varepsilon)^4}\right)^{k-k_0} \Delta_{k_0} \quad \forall k \geq k_0. \tag{17}$$

**(ii)** *(Non-asymptotic linear convergence) Let* $D_0 = \text{span}(\boldsymbol{\alpha}_0 - \boldsymbol{\alpha}^*) := \max(\boldsymbol{\alpha}_0 - \boldsymbol{\alpha}^*) - \min(\boldsymbol{\alpha}_0 - \boldsymbol{\alpha}^*)$. *If*

$$\phi(A_\varepsilon) + D_0 \leq \boldsymbol{\alpha}^* \oplus \boldsymbol{\beta}^* \leq \phi(B_\varepsilon) - D_0 \tag{18}$$

*for some* $\varepsilon > 0$, *then* (16) *holds with* $k_0 = 1$.

Part **(i)** states that whenever SSB $\mathbf{Z}^{\mathbf{r},\mathbf{c}}$ exists so that the problem is well-defined, the generalized Sinkhorn iterates (6) converges linearly *asymptotically*, meaning that the linear convergence kicks in after some unknown number $k_0$ of iterations. The linear rate of convergence is degraded if the condition number $\kappa$ for the cost matrix $\mathbf{C}$ is large. Part **(ii)** states in fact the linear convergence holds from the very first iteration if the $L^\infty$-ball around $\boldsymbol{\alpha}^* \oplus \boldsymbol{\beta}^*$ with radius $D_0$ takes entries from the domain of $\psi$, $(\phi(A), \phi(B))$. This holds if $\boldsymbol{\alpha}_0$ is sufficiently close to $\boldsymbol{\alpha}^*$ up to translation.

A sufficient condition for $\varepsilon > 0$ to satisfy (18) with zero initialization (by shifting the potentials appropriately so that $\|\boldsymbol{\alpha}^*\|_\infty \leq (\phi(B_\delta) - \phi(A_\delta))/2$) is

$$\phi(A_\varepsilon) \leq 2\phi(A_\delta) - \phi(B_\delta), \quad 2\phi(B_\delta) - \phi(A_\delta) \leq \phi(B_\varepsilon). \tag{19}$$

One can always find small $\varepsilon > 0$ satisfying the above provided $\phi(A) < 2\phi(A_\delta) - \phi(B_\delta)$ and $2\phi(B_\delta) - \phi(A_\delta) < \phi(B)$. The latter conditions always hold if $(\phi(A), \phi(B)) = \mathbb{R}$, which is the case for entropic optimal transport.

### 2.3. A priori bound on potentials

Our last concern is to obtain *dimension-independent* linear convergence rate of the generalized Sinkhorn iterates. This is crucial in computing generalized SSBs for large-scale problems using the proposed Sinkhon algorithm. The contraction rate in 2.6 is determined by $\sigma_\pm(\varepsilon)$, which are essentially determined by the tameness parameter $\delta$ for the

target margin $(\mathbf{r}, \mathbf{c})$. Hence, we only need to show that such $\delta$ is dimension-independent.

As it turns out, one can easily create a sequence of margins for which the tameness parameter vanishes at an arbitrary speed as the dimensions $m, n \to \infty$. In fact, determining whether a given margin $(\mathbf{r}, \mathbf{c})$ is $\delta$-tame for some dimension-independent $\delta$ is an important problem in entropic optimal transport (Marino & Gerolin, 2020; Carlier, 2022), random graphs with given degree sequences (Chatterjee et al., 2011; Barvinok & Hartigan, 2013; Dhara & Sen, 2022), enumeration of graphs with given degree sequences (Barvinok & Hartigan, 2013) and contingency tables (Barvinok & Hartigan, 2012; Lyu & Pak, 2022; Dittmer et al., 2020), and random matrices with given margins (Lyu & Mukherjee, 2024).

In what follows, we will provide some sufficient conditions for dimension-independent tameness. We will use easily-checked conditions on the margin of the following form: there exists some positive $s < t$ such that the target margin $(\mathbf{r}, \mathbf{c})$ satisfies the linear bounds

$$s \le \frac{\mathbf{r}(i)}{\mathbf{W}_{i\bullet}} \le t \quad \text{and} \quad s \le \frac{\mathbf{c}(j)}{\mathbf{W}_{\bullet j}} \le t \quad \text{for all } i, j. \quad (20)$$

We then seek for conditions on $s$ and $t$ such that $(\mathbf{r}, \mathbf{c})$ is $\delta$-tame for some $\delta$ that depends only on the divergence $f$ and $s, t$. It is hard to seek a simple condition that covers all possible range of problems entailed in the generalized SSB problem as the problem of tameness is known to depend very sensitively on the nature of the problem in the literature above. We will provide two such results depending on whether $f$ assumes bounded or unbounded domain.

We first consider the case when $f$ has bounded domain. Without loss of generality, we can then assume $\text{Interior}(\text{dom}(f)) = (0, B)$ for some positive finite $B$.

**Theorem 2.7** (Bound on potentials when $f$ has bounded domain)**.** *Suppose $A = 0$ and $B \in (0, \infty)$ and let $\mathbf{W}$ and $\kappa$ be as above. Fix constants $0 < s < t$ such that*

$$\kappa t < \sqrt{2Bs/\kappa} - s/\kappa \quad \text{or} \quad (\kappa^{-1}s + \kappa t)^2 < 4B\kappa^{-1}s. \quad (21)$$

*Then there exists a constant $C = C(f, s, t)$ with the following property: If any margin $(\mathbf{r}, \mathbf{c})$ satisfies (20) and $\mathcal{T}_{\mathbf{W}}(\mathbf{r}, \mathbf{c}) \cap (A, B)^{m \times n}$ is nonempty, then there exists a potential $(\boldsymbol{\alpha}, \boldsymbol{\beta})$ for margin $(\mathbf{r}, \mathbf{c})$ w.r.t. $\mathbf{W}$ such that $\|\boldsymbol{\alpha}\|_{\infty} \le C$ and $\|\boldsymbol{\beta}\|_{\infty} \le C$. In particular, $(\mathbf{r}, \mathbf{c})$ is $\delta$-tame for some $\delta > 0$ depending only on $f, \kappa, s$, and $t$.*

A few remarks are in order. First, the above result generalizes a similar result in (Lyu & Mukherjee, 2024) when $\mathbf{W} = \mathbf{1}_m \mathbf{1}_n^\top$ and $f$ is the KL-divergence between a compactly supported probability measure and its exponential tilting. Second, the result in Theorem 2.7 above cannot be

improved since the specialized result in (Lyu & Mukherjee, 2024) is already known to be *sharp*. This means that there is a sequence of margins with growing dimension satisfying the strict reverse inequality in (21) with vanishing tameness parameter. Third, the set of possible $(s, t)$ satisfying the condition (21) shrinks as the condition number $\kappa$ increases. Intuitively, it should be harder to guarantee dimension-independent tameness for wildly varying cost matrix $\mathbf{C}$. See Fig. 1.

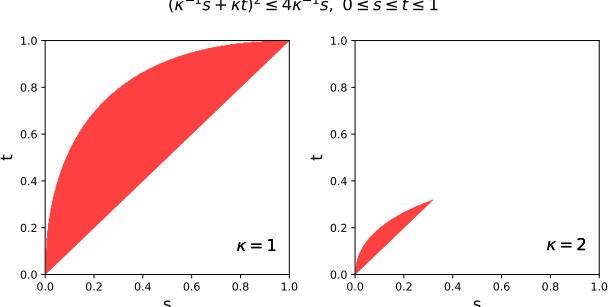

*Figure 1.* Red regions depict the solution set of the tameness condition (21) in Thm. 2.7 for $B = 1$ and $\kappa = 1, 2$.

Lastly, we consider the case when $f$ has unbounded domain. For instance, if $f(x) = x \log x$ as in Entropic Optimal Transport (EOT), then $(A, B) = (0, \infty)$ and $\psi(\theta) = e^\theta$ so $(f'(A), f'(B)) = \mathbb{R}$. The following theorem gives a sufficient condition for tameness for similar but more general cases.

**Theorem 2.8** (Bound on potentials when $f$ has unbounded domain)**.** *Suppose $(A, B) = (0, \infty)$ and $(0, \infty) \subseteq (f'(A), f'(B))$. Let $\mathbf{W}$ and $\kappa$ be as before. Fix constants $s, t \in (0, \infty)$ with $s \le t$. Let $(\mathbf{r}, \mathbf{c})$ be a $m \times n$ margin such that $\mathcal{T}_{\mathbf{W}}(\mathbf{r}, \mathbf{c}) \cap (A, B)^{m \times n}$ is nonempty and satisfies (20). Let $(\boldsymbol{\alpha}, \boldsymbol{\beta})$ be a potential for $(\mathbf{r}, \mathbf{c})$ w.r.t. $\mathbf{W}$. Denote*

$$c(\varepsilon; s, t) := \phi\left(\frac{\kappa^2 t(t - (1 - \varepsilon)s)}{\varepsilon s}\right) - \phi((1 - \varepsilon)t). \quad (22)$$

*Then entrywise,*

$$\phi(s) - \inf_{\varepsilon \in (0,1)} c(\varepsilon; s, t) \le \boldsymbol{\alpha} \oplus \boldsymbol{\beta} \le \phi(t) + \inf_{\varepsilon \in (0,1)} c(\varepsilon; s, t). \quad (23)$$

*In particular, $(\mathbf{r}, \mathbf{c})$ is $\delta$-tame for some $\delta > 0$ depending only on $f, \kappa, s$, and $t$ if the left-hand side of (23) is strictly larger than $f'(A)$.*

Consider again EOT, where $(\phi(A), \phi(B)) = \mathbb{R}$ and $\phi(\theta) = 1 + \log \theta$. In this case, the infimum in (23) is attained at $\varepsilon^* = \frac{s - t + \sqrt{t(t-s)}}{s}$, which yields

$$1 - 2\log\kappa + 2\log s - 2\log(\sqrt{t} + \sqrt{t-s})$$
$$\le \boldsymbol{\alpha} \oplus \boldsymbol{\beta} \le 1 + 2\log\kappa + \log t - \log s + 2\log(\sqrt{t} + \sqrt{t-s}).$$

In entropic optimal transport, one takes (see (3)),

$$\mathbf{C}_{ij} = \exp(-c(i,j)/\varepsilon) \qquad (24)$$

where $\varepsilon > 0$ now is the entropic regularization parameter. Then

$$\kappa = \exp\left(\varepsilon^{-1}(\max c(\cdot,\cdot) - \min c(\cdot,\cdot))\right) \le \exp(\|c\|_\infty/\varepsilon). \quad (25)$$

So our bound on the potentials above says

$$\boldsymbol{\alpha} \oplus \boldsymbol{\beta} = O(\|c\|_\infty/\varepsilon) + O(\log(t/s)). \qquad (26)$$

The corresponding bound on potentials for EOT in (Marino & Gerolin, 2020) and (Carlier, 2022) depend only on $\|c\|_\infty/\varepsilon$ and there is no dependence on the margin $(\mathbf{r}, \mathbf{c})$. This is because such bounds assume specific structure of the divergence (e.g., $f(x) = x\log x$) and $\mathbf{u}\mathbf{v}^\top = \mathbf{r}\mathbf{c}^\top$ as in (3). Our bound above does not assume $\mathbf{u}\mathbf{v}^\top = \mathbf{r}\mathbf{c}^\top$. Furthermore, our general bound in (23) works for general divergence $f$.

## 3. Analysis

Here we give high-level sketches of our analysis. Some of the proofs are relegated to Appendix B.

### 3.1. Generalized Kantorovich duality

For each $(\boldsymbol{\alpha}, \boldsymbol{\beta}) \in \mathbb{R}^m \times \mathbb{R}^n$, define

$$\mathbf{X}^{\boldsymbol{\alpha},\boldsymbol{\beta}} := \psi'(\boldsymbol{\alpha} \oplus \boldsymbol{\beta}). \qquad (27)$$

Using the key relation (9) and denoting $\widetilde{\mathbf{X}} = (\widetilde{\mathbf{X}}_{ij})$ with $\widetilde{\mathbf{X}}_{ij} := \mathbf{X}_{ij}\mathbf{W}_{ij}$, we can relate the primal objective function for SSB in (1) evaluated at $\mathbf{X}^{\boldsymbol{\alpha},\boldsymbol{\beta}}$ to the dual objective in (10) evaluated at $(\boldsymbol{\alpha}, \boldsymbol{\beta})$ as follows:

$$g(\mathbf{X}^{\boldsymbol{\alpha},\boldsymbol{\beta}}) = \sum_{i=1}^m \sum_{j=1}^n \left(\mathbf{X}_{ij}\phi(\mathbf{X}_{ij}) - \psi(\phi(\mathbf{X}_{ij}))\right)\mathbf{W}_{ij}$$

$$= \sum_{i=1}^m \sum_{j=1}^n (\boldsymbol{\alpha}(i) + \boldsymbol{\beta}(j))\widetilde{\mathbf{X}}_{ij} - \sum_{i,j} \psi(\phi(\mathbf{X}_{ij}))\mathbf{W}_{ij}$$

$$= \sum_{i=1}^m \boldsymbol{\alpha}(i) \sum_{j=1}^n \widetilde{\mathbf{X}}_{ij} + \sum_{j=1}^n \boldsymbol{\beta}(j) \sum_{i=1}^m \widetilde{\mathbf{X}}_{ij} - \sum_{i,j} \psi(\boldsymbol{\alpha}(i) + \boldsymbol{\beta}(j))\mathbf{W}_{ij}$$

$$= g^{\mathbf{r},\mathbf{c}}(\boldsymbol{\alpha}, \boldsymbol{\beta}) - \sum_{i=1}^m \boldsymbol{\alpha}(i)\left(\mathbf{r}(i) - \sum_{j=1}^m \widetilde{\mathbf{X}}_{ij}\right)$$

$$\quad - \sum_{j=1}^n \boldsymbol{\beta}(j)\left(\mathbf{c}(j) - \sum_{i=1}^n \widetilde{\mathbf{X}}_{ij}\right), \qquad (28)$$

where $g^{\mathbf{r},\mathbf{c}}(\boldsymbol{\alpha}, \boldsymbol{\beta})$ is defined in (10). This gives a direct relation between the primal and the dual objectives. Note that if $\bar{\mathbf{W}}$ satisfies row and column margin $(\mathbf{r}, \mathbf{c})$, or equivalently, if $\mathbf{X}^{\boldsymbol{\alpha},\boldsymbol{\beta}}$ has $\mathbf{W}$-weighted margin $(\mathbf{r}, \mathbf{c})$, then the above identity yields $g(\mathbf{X}^{\boldsymbol{\alpha},\boldsymbol{\beta}}) = g^{\mathbf{r},\mathbf{c}}(\boldsymbol{\alpha}, \boldsymbol{\beta})$. This is the essence of our proof of generalized Kantorovich duality stated in Theorem 2.4. The remaining details of the proof are given below.

**Proof of Theorem 2.4.** First we show (i). By the hypothesis, the SSB for margin $(\mathbf{r}, \mathbf{c})$ uniquely exists, which we denote by $\mathbf{Z}^{\mathbf{r},\mathbf{c}}$. Since $f' = \phi$, we have $\nabla g(\mathbf{Z}) = \left(\phi(\mathbf{Z}_{ij})\mathbf{W}_{ij}\right)_{ij}$. Since $\phi$ is differentiable, we can apply the multivariate Lagrange multiplier method, to conclude the existence of dual variables $\boldsymbol{\alpha}^* \in \mathbb{R}^m$ and $\boldsymbol{\beta}^* \in \mathbb{R}^n$ such that $\phi(\mathbf{Z}_{ij}) = \boldsymbol{\alpha}_i^* + \boldsymbol{\beta}_j^*$ whenever $\mathbf{W}_{ij} > 0$. Hence $(\boldsymbol{\alpha}^*, \boldsymbol{\beta}^*)$ is a potential for $(\mathbf{r}, \mathbf{c})$. Since $\mathbf{Z}^{\mathbf{r},\mathbf{c}} \in \mathscr{T}_{\mathbf{W}}(\mathbf{r}, \mathbf{c})$, $(\boldsymbol{\alpha}^*, \boldsymbol{\beta}^*)$ satisfies the Schrödinger system (5), which is exactly the first-order optimality condition for the dual (10). Since the dual problem is a concave maximization problem, it implies that $(\boldsymbol{\alpha}^*, \boldsymbol{\beta}^*)$ is a global maximizer for it. This shows (i).

Now suppose $(\hat{\boldsymbol{\alpha}}, \hat{\boldsymbol{\beta}})$ is a solution for the dual problem (10). Then $X^{\hat{\boldsymbol{\alpha}},\hat{\boldsymbol{\beta}}} \in \mathscr{T}_{\mathbf{W}}(\mathbf{r}, \mathbf{c}) \cap (A, B)^{m\times n}$, so the above yields

$$g(\mathbf{X}^{\hat{\boldsymbol{\alpha}},\hat{\boldsymbol{\beta}}}) = g^{\mathbf{r},\mathbf{c}}(\hat{\boldsymbol{\alpha}}, \hat{\boldsymbol{\beta}}) = \sup_{\boldsymbol{\alpha},\boldsymbol{\beta}} g^{\mathbf{r},\mathbf{c}}(\boldsymbol{\alpha}, \boldsymbol{\beta}). \qquad (29)$$

Furthermore, by Lemma 2.3, the SSB for margin $(\mathbf{r}, \mathbf{c})$ uniquely exists, which we denote by $\mathbf{Z}^{\mathbf{r},\mathbf{c}}$. By part (i), there exists a potential $(\boldsymbol{\alpha}^*, \boldsymbol{\beta}^*)$ for margin $(\mathbf{r}, \mathbf{c})$ such that $\mathbf{Z}^{\mathbf{r},\mathbf{c}} = \mathbf{X}^{\boldsymbol{\alpha}^*,\boldsymbol{\beta}^*}$. Then by using (29) with $(\boldsymbol{\alpha}^*, \boldsymbol{\beta}^*)$ and $(\hat{\boldsymbol{\alpha}}, \hat{\boldsymbol{\beta}})$, it follows that

$$\sup_{\boldsymbol{\alpha},\boldsymbol{\beta}} g^{\mathbf{r},\mathbf{c}}(\boldsymbol{\alpha}, \boldsymbol{\beta}) = g(\mathbf{X}^{\boldsymbol{\alpha}^*,\boldsymbol{\beta}^*}) = g(\mathbf{Z}^{\mathbf{r},\mathbf{c}}) \le g(\mathbf{X}^{\hat{\boldsymbol{\alpha}},\hat{\boldsymbol{\beta}}})$$

$$= g^{\mathbf{r},\mathbf{c}}(\hat{\boldsymbol{\alpha}}, \hat{\boldsymbol{\beta}}) = \sup_{\boldsymbol{\alpha},\boldsymbol{\beta}} g^{\mathbf{r},\mathbf{c}}(\boldsymbol{\alpha}, \boldsymbol{\beta}).$$

Thus all terms that appear above must equal, verifying (13). Since $g$ is strictly convex, $\mathbf{Z}^{\mathbf{r},\mathbf{c}}$ must be its unique global minimizer over $\mathscr{T}_{\mathbf{W}}(\mathbf{r}, \mathbf{c})$. Since $\mathbf{X}^{\hat{\boldsymbol{\alpha}},\hat{\boldsymbol{\beta}}} \in \mathscr{T}_{\mathbf{W}}(\mathbf{r}, \mathbf{c})$, the above yields that $\mathbf{X}^{\hat{\boldsymbol{\alpha}},\hat{\boldsymbol{\beta}}} = \mathbf{Z}^{\mathbf{r},\mathbf{c}}$. □

### 3.2. Convergence of generalized Sinkhorn

Recall that the Kantorovich dual (10) is a concave maximization problem and the objective function is not strictly concave. For such problems, linear convergence of standard optimization algorithms is not usually expected unless one already knows that the trajectory of an algorithm will be confined in a compact set and has an access on the strong convexity parameter of the objective function restricted on that set. Hence, key difficulty in establishing convergence of the generalized Sinkhorn algorithm 6 is showing that the sequence of dual variables along the trajectory of the Sinkhorn algorithm stays bounded. In the special case of entropic optimal transport, a uniform bound on the norm of the dual variables (e.g., (Marino & Gerolin, 2020)) is established relying heavily on the closed form of Sinkhorn iterates (7), which is enjoyed only for the special case. In the lemma below, we establish that the generalized Sinkhorn algorithm is a non-expanding operator in the $L^\infty$-norm.

**Lemma 3.1** ($L^\infty$-monotonicity of the Sinkhorn iterates)**.** *Suppose* $\mathscr{T}_{\mathbf{W}}(\mathbf{r}, \mathbf{c}) \cap (A, B)^{m\times n}$ *is non-empty. Let* $(\boldsymbol{\alpha}_t, \boldsymbol{\beta}_t)$,

$t \geq 0$ *denote the iterates produced by the Sinkhorn algorithm* (6). *Let* $(\hat{\boldsymbol{\alpha}}, \hat{\boldsymbol{\beta}})$ *be an arbitrary potential for the margin* $(\mathbf{r}, \mathbf{c})$ *w.r.t.* $\mathbf{W}$. *Then for each* $t_0 \geq 0$,

$$\sup_{t \geq t_0} \|(\boldsymbol{\alpha}_t, \boldsymbol{\beta}_t) - (\hat{\boldsymbol{\alpha}}, \hat{\boldsymbol{\beta}})\|_\infty \leq \|\boldsymbol{\alpha}_{t_0} - \hat{\boldsymbol{\alpha}}\|_\infty. \quad (30)$$

A special case of the above lemmas was first obtained by (Lyu & Mukherjee, 2024) for analyzing Sinkhorn algorithm for typical table problems arising from random matrix with given margin. Their key idea was to realize that the Jacobian matrix of the mappings $\boldsymbol{\alpha}_{k-1} \mapsto \boldsymbol{\beta}_k$ and $\boldsymbol{\beta}_k \mapsto \boldsymbol{\alpha}_k$ are row-stochastic matrices by using the implicit function theorem rather than using a closed-form expression of these updates, which is only available for special cases such as EOT. Our proof of Lemma 3.1 above follows their approach.

Note that by shift invariance of potentials, the bound (30) in Lemma 3.1 continues to hold if we replace the row potential $\hat{\boldsymbol{\alpha}}$ with any translation $\hat{\boldsymbol{\alpha}} + \lambda \mathbf{1}_m$. Hence the right-hand side of (30) takes the minimum value of $2D_0$, where $D_0$ is defined in Thm. 2.6 **(ii)**. This means that all future Sinkhorn iterates $(\boldsymbol{\alpha}_t, \boldsymbol{\beta}_t)$ stay within the $L_\infty$ neighborhood of $(\hat{\boldsymbol{\alpha}}, \hat{\boldsymbol{\beta}})$ with side length $2D_0$. If such a box is contained in the box $(\phi(A_\delta), \phi(B_\delta))^{(m+n)}$ for some $\delta > 0$, this would imply that every intermediate potential $(\boldsymbol{\alpha}_t, \boldsymbol{\beta}_t)$ is $\delta$-tame. When $(\phi(A_\delta), \phi(B_\delta)) = \mathbb{R}$ (e.g., for EOT when $\phi(x) = f'(x) = 1 + \log x$ and $A = 0$, $B = \infty$), this is always the case. However, one has to pay additional care when $(\phi(A_\delta), \phi(B_\delta))$ is not the entire real line (e.g., it is $(-\infty, 0)$ when $f(x) = x \log x + (1-x) \log(1-x)$ for random graphs with given degree sequence).

Once we establish a priori bounds on the generalized Sinkhorn iterates, we can analyze linear convergnece of the iterates by following the appraoch of Carlier (2022) for EOT, which was recently generalized by (Lyu & Mukherjee, 2024) for the generalized typical table problem. We adopt their approach to prove Theorem 2.6. Essentially, the idea is to use the strong convexity and smoothness of the entrywise dual objective $\psi$ (see (10)) within a compact interval. This interval comes from the a priori bound of Sinkhorn iterates given in Lemma 3.1. Using this, one can show that the algorithm is a contraction in the space of direct-sum potentials $\boldsymbol{\alpha} \oplus \boldsymbol{\beta}$, equipped with the weighted norm $\|\cdot\|_{\mathbf{uv}^\top}$ introduced above the statement of Thm. 2.6. Since linear convergence occurs at the level of direct-sum potentials, this does not contradict the fact that there are infinitely many solutions to the Kantorovich dual (10) coming from the shift-invariance of potentials.

### 3.3. A priori bounds on the potentials

In our theory of generalized SSB, obtaining a priori bounds on the potentials requires a substantial innovation from the existing literature of EOT and typical tables. Unlike in EOT, we cannot rely on separability of dual objective, i.e., $\psi'(\alpha + \beta) = \psi'(\alpha)\psi'(\beta)$ (since $\psi'(x) = \exp(x)$), and unlike in the typical table problem with uniform weight $\mathbf{W} = \mathbf{1}_m \mathbf{1}_n^\top$ (with constant cost), we cannot use reduction techniques developed in (Lyu & Mukherjee, 2024) that rely on the symmetry of the weights.

Theorem 2.7 deals with the case when $\text{dom}(f)$ is bounded. Without loss of generality assume $\boldsymbol{\alpha}(1) \leq \cdots \leq \boldsymbol{\alpha}(m)$ and $\boldsymbol{\beta}(1) \leq \cdots \leq \boldsymbol{\beta}(n)$. In order to arrive at a dimension independent sufficient condition we analyze how different shifts in the potentials (using the shift invariance property) can impact the relative contributions of vectors $\mathbf{u}$ and $\mathbf{v}$ in arriving at a pair of constants $0 < s < t$ which satisfy condition (21). Once (21) is satisfied, it is easy to find a constant C, using $\phi$ is continuous and increasing , such that $\|\boldsymbol{\alpha}\|_\infty \leq C$ and $\|\boldsymbol{\beta}\|_\infty \leq C$ which readily translates to $\delta$-tameness for some $\delta > 0$ that depends on $C$ and $\kappa$. Since $C$ itself depends on $f, s$, and $t$, we get that $\delta$ depends on $f, \kappa, s$, and $t$.

Theorem 2.8 deals with the unbounded case. The first step is to bound the largest coordinate of the column potential by $\boldsymbol{\beta}(n) \leq \phi(t)$. Indeed,

$$t \mathbf{W}_{\bullet n} \geq \mathbf{c}(n) = \sum_{i=1}^{m} \mathbf{W}_{in} \psi'(\boldsymbol{\alpha}(i) + \boldsymbol{\beta}(n))$$
$$\geq \sum_{i=1}^{m} \mathbf{W}_{in} \psi'(\boldsymbol{\beta}(n)) = \psi'(\boldsymbol{\beta}(n)) \mathbf{W}_{\bullet n}.$$

Next, we look at the smallest row sum $\mathbf{r}(1)$. The key insight here is to break the weighted sum for $\mathbf{r}(1)$ depending on whether $\boldsymbol{\beta}(j)$ is smaller than $\phi((1-\lambda)s)$. This is because if most of $\boldsymbol{\beta}(j)$'s are less than this value, then the first row sum will be too small and it will violate the lower bound on $\mathbf{r}(1)$ in terms of the quantity $s$. Proceeding this way,

$$s \mathbf{W}_{1\bullet} \leq \mathbf{r}(1) = \sum_{\boldsymbol{\beta}(j) < \phi((1-\lambda)s)} \mathbf{W}_{1j} \psi'(\boldsymbol{\beta}(j))$$
$$+ \sum_{\boldsymbol{\beta}(j) \geq \phi((1-\lambda)s)} \mathbf{W}_{1j} \psi'(\boldsymbol{\beta}(j))$$
$$\leq M(t - (1-\lambda)s) + \mathbf{W}_{1\bullet}(1-\lambda)s,$$

where we denote $M := \sum_{j: \boldsymbol{\beta}(j) \geq \phi((1-\lambda)s)} \mathbf{W}_{1j}$. Rearranging terms and since $\mathbf{W}_{1\bullet}, s > 0$, we deduce

$$\frac{M}{\mathbf{W}_{1\bullet}} \geq \frac{\lambda s}{t - (1-\lambda)s} > 0. \quad (31)$$

Thus a linear fraction of $j$'s satisfy $\boldsymbol{\beta}(j) \geq \phi((1-\lambda))s)$. Using this fact, this can be used to upper bound $\boldsymbol{\alpha}(m)$. Since both the row and column potentials are now upper bounded and since the row and column sums are lower bounded, the smallest row and column potentials now cannot be too small. This leads to lower bounds on $\boldsymbol{\alpha}(1)$ and $\boldsymbol{\beta}(1)$.

## 4. Numerical Experiments

In this section, we present results on the effectiveness of our GSA (6) for various strictly convex divergences. We consider the following six strictly convex f-divergences:

- KL divergence: $f(x) = x \log x$,
- Jeffreys divergence: $f(x) = (x-1) \log x$,
- Jensen-Shannon divergence: $f(x) = \frac{1}{2}(x \log x - (x+1) \log(\frac{x+1}{2}))$,
- Neyman $\chi^2$ divergence: $f(x) = (x-1)^2$,
- Binomial divergence: $f(x) = 10 \log(\frac{1+e^x}{2})$, and
- Squared Hellinger divergence: $f(x) = \frac{1}{2}(\sqrt{x}-1)^2$.

For different combinations of the (source and target) marginals and the cost matrix, we plot the log-scale improvement in the $L^1$ errors of the GSA iterates of the f-divergences above versus the iteration count. We highlight two such combinations here, in Fig. 2 and Fig. 3. For the sake of brevity, we plot only the resultant transportation map given by Jeffrey's divergence, while covering all remaining transportation maps in Appendix C.

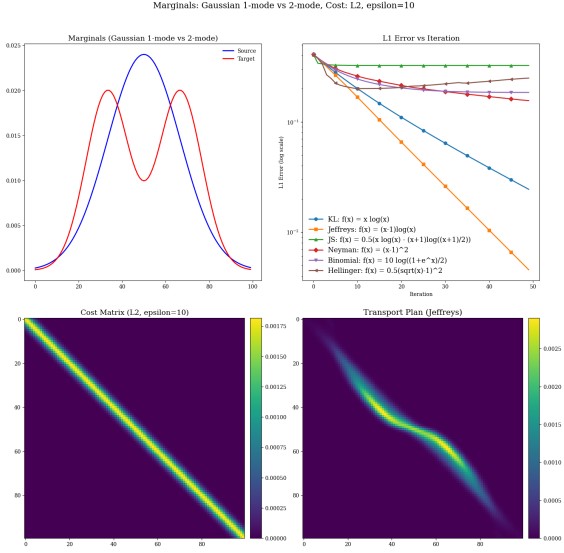

*Figure 2.* Convergence of dual objective gradient norms of GSSB, transportation maps for various f-divergences for Gaussian marginals in dimensions $m = 100$ and $n = 100$, $L^2$ cost function, and entropic regularization parameter $\epsilon = 10$. The marginals are shown in the subplot (1,1), the cost matrix $\mathbf{C}$ is shown at (2,1), and the $L^1$ gradient norm of the dual objective (also the $L^1$ marginal error) is shown at (1,2). Plot (2,2) shows the transport map given by Jeffrey's divergence. For a more thorough transport map comparison, we direct the reader to Figure 8.

We observe that using different divergence functions shows different linear convergence rates; sometimes, the standard KL divergence is not the fastest one. In addition, the transport maps in Appendix C show qualitative differences depending on the divergence. This indicates the potential

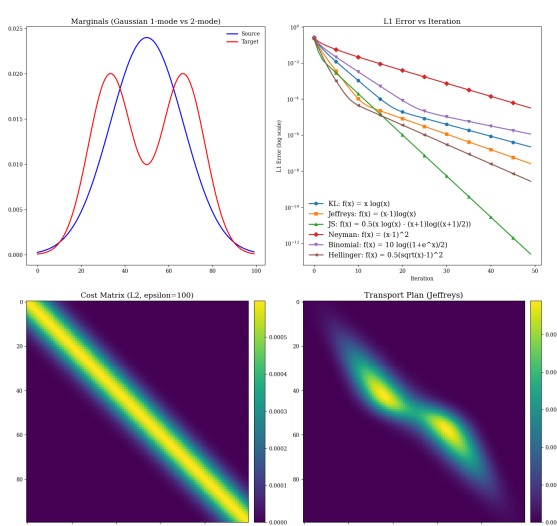

*Figure 3.* Convergence of dual objective gradient norms of GSSB, transportation maps for various f-divergences for Gaussian marginals in dimensions $m = 100$ and $n = 100$, $L^2$ cost function, and entropic regularization parameter $\epsilon = 10$. The marginals are shown in the subplot (1,1), the cost matrix $\mathbf{C}$ is shown at (2,1), and the $L^1$ gradient norm of the dual objective (also the $L^1$ marginal error) is shown at (1,2). Plot (2,2) shows the transport map given by Jeffrey's divergence. For a more thorough transport map comparison, we direct the reader to Figure 9.

benefit of using more adapted divergence functions in EOT problem formulations. We postpone a more detailed exposition of our numerical experiments to Appendix C.

## 5. Conclusion

In this paper, we present the generalized static Schrödinger bridge problem, establish its Kantorovich dual, and show that the associated generalized Sinkhorn algorithm converges linearly in a dimension independent manner under mild assumptions on the general divergence functional $f$, weight matrix $\mathbf{W}$ and margin $(\mathbf{r}, \mathbf{c})$. We now discuss some immediate open problems. One question of particular interest is if our analysis, which crucially uses a centering argument (possible by shift-invariance of dual potentials) can be extended to the more general problem considered in (Luo & Tseng, 1992) of the form

$$\min_{\mathbf{x}} \quad f(\mathbf{x}) := g(\mathbf{Ex}) + \langle \mathbf{b}, \mathbf{x} \rangle \quad s.t. \quad \mathbf{x} \in \mathbf{X},$$

where $g$ is strictly convex to achieve non-asymptotic linear convergence for alternating minimization and even coordinate descent methods under milder assumptions than already known. Another key question is whether the a priori bounds on potentials can be strengthened to sharp conditions such as those for Barvinok margins.

## Impact statement

This paper presents work whose goal is to advance the field of Machine Learning. There are many potential societal consequences of our work, none of which we feel must be specifically highlighted here.

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

# Linear Convergence of Sinkhorn's Algorithm for Generalized Static Schrödinger Bridge
## Supplementary Material

## A. Background

**Connection to static Schrödinger Bridge.** Our prime motivation to study the problem (1) is the classical *static Schrödinger bridge*. Suppose we have a probability measure $\mathscr{R}$ on $\mathscr{X} \times \mathscr{X}$ representing our prior knowledge on the joint behavior of two $\mathscr{X}$-valued random variables, say $X$ and $Y$. Suppose that we are given marginal distributions of $X$ and $Y$, say $\mu_X$ and $\mu_Y$, respectively. The static Schrödinger bridge in this setting is to find the 'most natural' joint probability measure $\mathscr{H}$ for $(X, Y)$ such that (1) it has the correct marginal distributions $(\mu_X, \mu_Y)$ and (2) it is as close as possible to the prior distribution $\mathscr{R}$.

More precisely, given two probability measures $\lambda, \nu$ on the same measureable space, define *the relative entropy* (or the Kullback–Leibler (KL) divergence) of $\mu$ from $\nu$ by

$$D_{KL}(\lambda \,\|\, \nu) := \begin{cases} \int \frac{d\lambda}{d\nu}(u) \log\left(\frac{d\lambda}{d\nu}(u)\right) \nu(du) & \text{if } \lambda \ll \nu \\ \infty & \text{otherwise,} \end{cases} \tag{32}$$

where $\lambda \ll \nu$ means that $\lambda$ is absolutely continuous w.r.t. $\nu$ and $\frac{d\lambda}{d\nu}$ denotes the Radon-Nikodym derivative of $\lambda$ with respect to $\nu$. Now the static Schrödinger bridge problem introduced informally above can be stated as

$$\min_{\mathscr{H} \in \Pi(\mu_X, \mu_Y)} D_{KL}(\mathscr{H} \,\|\, \mathscr{R}), \tag{33}$$

where $\Pi(\mu_X, \mu_Y)$ denotes the set of all joint distributions in $\mathscr{X} \times \mathscr{X}$ that has marginal distributions $\mu_X$ and $\mu_Y$. The joint probability measures $\mathscr{H}$ that solves the above problem is called the *static Schrödinger bridge between marginal distributions $\mu_X$ and $\mu_Y$ w.r.t. $\mathscr{R}$*.

The base model is often taken to be the product measure $\mathscr{R} = \mu_1 \otimes \mu_2$. In this case, there exists functions $\boldsymbol{\alpha}_1, \boldsymbol{\alpha}_2 : \mathbb{R} \to \mathbb{R}$ known as the *Schrödinger potentials* that characterize the Schrödinger bridge as (see, e.g., (Léonard, 2013; Nutz & Wiesel, 2022) and Sec. 3 (B) of (Csiszár, 1975))

$$\frac{d\mathscr{H}}{d\mathscr{R}}(x, y) = e^{\boldsymbol{\alpha}_1(x) + \boldsymbol{\alpha}_2(y)} \qquad \mathscr{R}\text{-a.s.} \tag{34}$$

To make the connection between our main problem (1) and the static Schrödinger bridge (33) clear, consider the following discrete setting where the random variables $X$ and $Y$ take $m$ and $n$ distinct values, respectively. We will identify them with the integers in $[m] := \{1, \ldots, m\}$ and $[n]$, respectively. Then the prior measure $\mathscr{R}$ lives on the integer lattice $[m] \times [n]$ and the marginal distributions $\mu_X$ and $\mu_Y$ live on $[m]$ and $[n]$, respectively. Then writing (33) as an optimization problem involving the Radon-Nikodym derivative $\mathbf{X} = \frac{d\mathscr{H}}{d\mathscr{R}}$, we obtain

$$\min_{\mathbf{X} = (x_{ij}) \in \mathbb{R}_{>0}^{m \times n}} \sum_{i, j} (x_{ij} \log x_{ij}) \mathscr{R}(i, j) \tag{35}$$

$$\text{s.t. } \sum_{j=1}^{n} x_{ij} \mathscr{R}(i, j) = \mu_X(i), \ \sum_{i=1}^{m} x_{ij} \mathscr{R}(i, j) = \mu_Y(j) \ \forall i, j.$$

Notice that the above is a special case of (1) with weight matrix $\mathbf{W} = \mathscr{R}$, margin $(\mu_X, \mu_Y)$, and divergence $f(x) = x \log x$.

**Connection to Entropic Optimal Transport.** Schrödinger bridges and entropic optimal transport (EOT) are intimately related as evidenced by the problem formulation that arises in EOT. More precisely, given a measurable cost function $c : \mathscr{X} \times \mathscr{X} \to \mathbb{R}_{\geq 0}$ and a fixed regularization parameter $\varepsilon > 0$, the entropic optimal transport problem is

$$\min_{\mathscr{H} \in \Pi(\mu_X, \mu_Y)} \int c \, d\mathscr{H} + \varepsilon D_{KL}(\mathscr{H} \| \mu_X \otimes \mu_Y) \tag{36}$$

which is the same as (33) for $\mathscr{R}(d(x,y)) \propto e^{-c(x,y)/\varepsilon} \mu_X(dx) \otimes \mu_Y(dy)$. We differentiate between the cost matrix $c_{ij}$ and the margin $\mathbf{c}$ by boldface. In a manner similar to earlier, consider the discrete case and denote the Radon-Nikodym derivative as $\mathbf{X} = \frac{d\mathscr{H}}{d\mathscr{R}}$. Then (35) further reduces to

$$\min_{\mathbf{X}=(x_{ij})\in\mathbb{R}^{m\times n}_{>0}} \sum_{i,j} (x_{ij}\log x_{ij}) e^{-c_{ij}/\varepsilon} \mu_X(i)\mu_Y(j) \tag{37}$$

$$\text{s.t. } \sum_{j=1}^{n} x_{ij} e^{-c_{ij}/\varepsilon} \mu_Y(j) = 1, \ \sum_{i=1}^{m} x_{ij} e^{-c_{ij}/\varepsilon} \mu_X(i) = 1 \ \forall i,j.$$

The above may look a bit different from a direct discretization of the standard formulation of the EOT (36). The difference is that while (36) is formulated in temrs of the joint coupling $\mathscr{H}$ as the optimization variable, (37) is formulated in terms of the Radon-Nikodym drivative $\mathbf{X} = \frac{d\mathscr{H}}{d\mathscr{R}}$. Recall that (37) is a special case of our main problem (1) with weight $\mathbf{W}_{ij} \propto (e^{-c_{ij}/\varepsilon} \mu_X(i)\mu_Y(j))_{ij}$, margin $(\mathbf{r},\mathbf{c}) = (\mu_X, \mu_Y)$ and divergence $f(x) = x\log x$.

**Connection to Random matrices with given margins.** Recently, Lyu and Mukherjee (Lyu & Mukherjee, 2024) discovered that the random matrix with i.i.d. entries from a common distribution $\mu$ *conditioned* to have a prescribed row margin $\mathbf{r} \in \mathbb{R}^m$ and column margin $\mathbf{c} \in \mathbb{R}^n$ concentrates around the solution of a special case of (1). Specifically, it is the case when $\mathbf{W} = \mathbf{1}\mathbf{1}^\top$ (uniform prior) and $f(x) = D_{KL}(\mu_{\phi(x)} \| \mu)$, where $\mu_\theta$ is the exponential tilt of $\mu$ defined by $\frac{d\mu_\theta}{d\mu}(x) = \exp(\theta x - \psi(\theta))$, $\psi(\theta) = \log \int e^{\theta x} \mu(dx)$ is the log-Laplace transform of $\mu$, and $\phi$ is the strictly increasing inverse of the 'tilt-to-mean' function $\psi'$.

The key insight in (Lyu & Mukherjee, 2024) is that the margin-conditioned random matrix $X$ can be well-approximated by another random matrix $Y$ with independent entries, where the laws of the entries are suitable exponential tilt of the base measure $\mu$. The authors show that among such parameterized models by entrywise exponential tilts, the best one is given by a rank-one tilting, where $Y_{ij} \sim \mu_{\alpha(i)+\beta(j)}$ for some vectors $\boldsymbol{\alpha} \in \mathbb{R}^m$ and $\boldsymbol{\beta} \in \mathbb{R}^n$. The best such dual variables can be computed by solving the maximum-likelihood estimation problem, which is precisely (10) with $\mathbf{W} = \mathbf{1}\mathbf{1}^\top$. Since the log-likelihood function $g^{\mathbf{r},\mathbf{c}}(\boldsymbol{\alpha},\boldsymbol{\beta})$ is strictly concave, the MLE must be a critical point, yielding the MLE equation

$$\mathbb{E}[Y] = \psi'(\boldsymbol{\alpha} \oplus \boldsymbol{\beta}) \in \mathscr{T}(\mathbf{r},\mathbf{c}). \tag{38}$$

Here, the equality follows from the definition and $\mathscr{T}(\mathbf{r},\mathbf{c}) = \mathscr{T}_{\mathbf{W}}(\mathbf{r},\mathbf{c})$ is the transportation polytope of all $m \times n$ real matrices with margin $(\mathbf{r},\mathbf{c})$ (recall $\mathbf{W} = \mathbf{1}\mathbf{1}^\top$). This consists of $m+n$ linear equations (one being redundant) that precisely correspond to the system of Schrödinger equations (see, e.g., (Nutz, 2021)).

Once the MLE $(\boldsymbol{\alpha}, \boldsymbol{\beta})$ is found, then the best approximating model $Y$ for $X$ is given by

$$Y \sim \mu_{\boldsymbol{\alpha} \oplus \boldsymbol{\beta}}, \tag{39}$$

meaning that $Y_{ij}$s are independent and $Y_{ij} \sim \mu_{\boldsymbol{\alpha}(i)+\boldsymbol{\beta}(j)}$. In (Lyu & Mukherjee, 2024), Lyu and Mukherjee established 'transference principles', which roughly say that $X \approx Y$. Given this, it is natural to expect that $X$ concentrates around the mean of $Y$, which is precisely the matrix $\psi'(\boldsymbol{\alpha} \oplus \boldsymbol{\beta})$. This matrix is known as the *typical table* for the margin $(\mathbf{r},\mathbf{c})$ and base measure $\mu$ in (Lyu & Mukherjee, 2024). According to our Theorem 2.4, the typical table can also be viewed as the corresponding generalized static Schrödinger bridge.

Much of the theories we develop in this work are in parallel to and generalize those in (Lyu & Mukherjee, 2024) for non-uniform weight matrix $\mathbf{W}$ and general divergence function $f(\cdot)$ that is not of the form $D_{KL}(\mu_{\phi(\cdot)} \| \mu)$.

# B. Postponed Proofs

Recall that we denote $\text{Interior}(\text{dom}(f)) = (A,B)$ for some $A, B \in \mathbb{R} \cup \{\infty\}$ and $\Theta^\circ := \text{Interior}(\text{dom}(\psi))$.

***Proof of Lemma** 2.3.* Since the domain of $f'$ is $(A,B)$, if a SSB $\mathbf{Z}^{\mathbf{r},\mathbf{c}}$ exists, it belongs to the intersection $\mathscr{T}_{\mathbf{W}}(\mathbf{r},\mathbf{c}) \cap (A,B)^{m\times n}$. For the other direction, suppose there exists some $\mathbf{X} = (x_{ij}) \in \cap(A,B)^{m\times n}$. Then there exists $\delta > 0$ such that $\mathbf{X} \in [A_\delta, B_\delta]^{m\times n}$ (see Def. 2.5). Let $\mathbf{Z}^{(k)}$ be a sequence of matrices in $(A,B)^{m\times n} \cap \mathscr{T}_{\mathbf{W}}(\mathbf{r},\mathbf{c})$ such that

$$g(\mathbf{Z}^{(k)}) \le \inf_{Z \in \mathscr{T}_{\mathbf{W}}(\mathbf{r},\mathbf{c}) \cap (A,B)^{m\times n}} g(\mathbf{Z}) + \frac{1}{k} \tag{40}$$

if the infimum is finite, and require $g(\mathbf{Z}^{(k)}) \geq k$ otherwise. We begin by showing that for any $i, j$ such that $\mathbf{W}_{ij} > 0$,

$$A < \liminf_{k \to \infty} \mathbf{Z}_{ij}^{(k)} \leq \limsup_{k \to \infty} \mathbf{Z}_{ij}^{(k)} < B. \tag{41}$$

By passing to a subsequence, assume that $\mathbf{Z}_{ij}^{(k)} \to \mathbf{Z}_{ij}^{(\infty)} \in [A, B]$ for all $ij$, where $\mathbf{Z}^{(\infty)} = (\mathbf{Z}_{ij}^{(\infty)})$ is a (possibly) extended real-valued matrix. Let

$$\mathscr{I}_A := \{(i, j) : \mathbf{Z}_{ij}^{(\infty)} = A, \mathbf{W}_{ij} > 0\} \tag{42}$$

$$\mathscr{I}_B = \{(i, j) : \mathbf{Z}_{ij}^{(\infty)} = B, \mathbf{W}_{ij} > 0\}, \tag{43}$$

$$\mathscr{I}_{A,B} := \{(i, j) : \mathbf{Z}_{ij}^{(\infty)} \in (A, B)\}. \tag{44}$$

For any $\lambda \in [0, 1]$ set $\mathbf{Z}^{(k,\lambda)} := (1 - \lambda)\mathbf{Z}^{(k)} + \lambda\mathbf{X}$, and note that $\mathbf{Z}^{(k,\lambda)} \in \mathscr{T}_\mathbf{W}(\mathbf{r}, \mathbf{c}) \cap [A_{\lambda\delta}, B_{\lambda\delta}]^{m \times n}$. Hence

$$g(\mathbf{Z}^{(k)}) \leq \inf_{\mathbf{X} \in \mathscr{T}_\mathbf{W}(\mathbf{r}, \mathbf{c}) \cap [A_{\lambda\delta}, B_{\lambda\delta}]^{m \times n}} g(\mathbf{X}) + \frac{1}{k} \leq g(\mathbf{Z}^{(k,\lambda)}) + \frac{1}{k} \tag{45}$$

for all sufficiently large $k \geq 1$. By convexity of $g$,

$$\frac{1}{k\lambda} \geq \frac{g(\mathbf{Z}^{(k)}) - g(\mathbf{Z}^{(k,\lambda)})}{\lambda} \geq \langle \nabla g(\mathbf{Z}^{(k,\lambda)}), \mathbf{W} \odot (\mathbf{Z}^{(k)} - \mathbf{X}) \rangle = \sum_{i,j} f'\left(\mathbf{Z}_{ij}^{(k,\lambda)}\right)(\mathbf{Z}_{ij}^{(k)} - x_{ij})\mathbf{W}_{ij}, \tag{46}$$

Letting $k \to \infty$ followed by $\lambda \searrow 0$ in (46) we get

$$0 \geq -\sum_{ij} \phi(\mathbf{Z}_{ij}^{(\infty)})(\mathbf{Z}_{ij}^{(\infty)} - x_{ij})\mathbf{W}_{ij} \tag{47}$$

$$= \sum_{(i,j) \in \mathscr{I}_A} \phi(A)(A - x_{ij})\mathbf{W}_{ij} + \sum_{(i,j) \in \mathscr{I}_B} \phi(B)(B - x_{ij})\mathbf{W}_{ij} + \sum_{(i,j) \in \mathscr{I}_{A,B}} \phi(\mathbf{Z}_{ij}^{(\infty)})(\mathbf{Z}_{ij}^{(\infty)} - x_{ij})\mathbf{W}_{ij}. \tag{48}$$

Note that $\mathscr{I}_A = \emptyset$ if $A = -\infty$ and $\phi(A) = -\infty$ if $A$ is finite by Assumption 2.1. Similarly, $\mathscr{I}_B = \emptyset$ if $B = \infty$ and $\phi(B) = \infty$ if $B$ is finite. Since the third term above is finite, the above equality holds only if $\mathscr{I}_A = \mathscr{I}_B = \emptyset$, which gives (41).

Given (41), we have the existence of $\delta > 0$ such that

$$\inf_{\mathbf{X} \in \mathscr{T}_\mathbf{W}(\mathbf{r}, \mathbf{c}) \cap (A, B)^{m \times n}} g(\mathbf{Z}) = \inf_{\mathbf{X} \in \mathscr{T}_\mathbf{W}(\mathbf{r}, \mathbf{c}) \cap [A_\delta, B_\delta]^{m \times n}} g(\mathbf{Z}). \tag{49}$$

But the RHS above focuses on a compact set, and minimizes a strictly convex function. Hence the existence of a unique optimizer follows. $\qquad\square$

***Proof of Lemma 3.1.*** By permuting the rows and columns if necessary, we may assume that $\hat{\boldsymbol{\alpha}}(1) \leq \cdots \leq \hat{\boldsymbol{\alpha}}(m)$ and $\hat{\boldsymbol{\beta}}(1) \leq \cdots \leq \hat{\boldsymbol{\beta}}(n)$. Fix $(\boldsymbol{\alpha}, \boldsymbol{\beta}) \in \mathbb{R}^m \times \mathbb{R}^n$. Let $\boldsymbol{\beta} \mapsto \xi(\boldsymbol{\beta}) =: \boldsymbol{\alpha}'$ denote the Sinkhorn update for the first dual variable given the second one $\boldsymbol{\beta}$. This update is characterized by (5) as $\mathbf{r}(i) = \sum_j \psi'(\boldsymbol{\alpha}'(i) + \boldsymbol{\beta}(j))\mathbf{W}_{ij}$ for all $i$. We would like to compute the Jacobian of this map. To do so, define the function $F : \mathbb{R}^m \times \mathbb{R}^n \to \mathbb{R}^m$ by setting the $i$th coordinate of $F(\boldsymbol{\alpha}, \boldsymbol{\beta}) \in \mathbb{R}^m$ for $i = 1, \ldots, m$ as

$$\mathbf{r}(i) - \sum_j \psi'(\boldsymbol{\alpha}(i) + \boldsymbol{\beta}(j))\mathbf{W}_{ij}. \tag{50}$$

Then $\boldsymbol{\alpha}' = \xi(\boldsymbol{\beta})$ is the unique zero of the equation $F(\cdot, \boldsymbol{\beta}) = \mathbf{0}$. Let $E = E(\boldsymbol{\alpha}', \boldsymbol{\beta})$ be the $m \times n$ matrix whose $(i, j)$ entry is $-\psi''(\boldsymbol{\alpha}'(i) + \boldsymbol{\beta}(j)')\mathbf{W}_{ij}$ and let $E_{i\bullet}$ denote the $i$th row sum of $E$ for $i = 1, \ldots, m$. Then the Jacobian of $F$ with respect to $\boldsymbol{\alpha}$ and $\boldsymbol{\beta}$, respectively, are given by

$$[J_{F;\boldsymbol{\alpha}}(\boldsymbol{\alpha}', \boldsymbol{\beta})]_{m \times m} = \operatorname{diag}(E_{1\bullet}, \ldots, E_{m\bullet}), \qquad [J_{F;\boldsymbol{\beta}}(\boldsymbol{\alpha}', \boldsymbol{\beta})]_{m \times n} = E. \tag{51}$$

The first Jacobian matrix above is always invertible since $\psi'' > 0$ on the domain. Hence by the implicit function theorem,

$$[J_{\boldsymbol{\alpha}';\boldsymbol{\beta}}]_{m \times n} = -[J_{F;\boldsymbol{\alpha}}(\boldsymbol{\alpha}', \boldsymbol{\beta})]_{m \times m}^{-1}[J_{F;\boldsymbol{\beta}}(\boldsymbol{\alpha}', \boldsymbol{\beta})]_{m \times n} \tag{52}$$

$$= -\left[E(\boldsymbol{\alpha}', \boldsymbol{\beta})_{ij} / E(\boldsymbol{\alpha}', \boldsymbol{\beta})_{i\bullet}\right]_{m \times n}. \tag{53}$$

Importantly, we observe that $-[J_{\boldsymbol{\alpha}';\boldsymbol{\beta}}]$ is a row-stochastic matrix.

Now fix any potential $(\hat{\boldsymbol{\alpha}}, \hat{\boldsymbol{\beta}})$ for the margin $(\mathbf{r}, \mathbf{c})$. Note that $\xi(\hat{\boldsymbol{\beta}}) = \hat{\boldsymbol{\alpha}}$. Let $\gamma(s) = (1-s)\boldsymbol{\beta} + s\hat{\boldsymbol{\beta}}$ denote the linear interpolation between $\boldsymbol{\beta}$ and $\hat{\boldsymbol{\beta}}$. Then denoting $P_s := -J_{\xi(\gamma(s));\gamma(s)}$, we have

$$\boldsymbol{\alpha}' - \hat{\boldsymbol{\alpha}} = \xi(\boldsymbol{\beta}) - \xi(\hat{\boldsymbol{\beta}}) = \underbrace{\left[ \int_0^1 P_s \, ds \right]}_{=:P} (\hat{\boldsymbol{\beta}} - \boldsymbol{\beta}). \tag{54}$$

The matrix $P$ defined above is row-stochastic since every intermediate negative Jacobian matrix is row-stochastic by the earlier observation. In particular, since $\|P\|_\infty = 1$, this yields

$$\|\boldsymbol{\alpha}' - \hat{\boldsymbol{\alpha}}\|_\infty \le \|P\|_\infty \|\hat{\boldsymbol{\beta}} - \boldsymbol{\beta}\|_\infty = \|\hat{\boldsymbol{\beta}} - \boldsymbol{\beta}\|_\infty. \tag{55}$$

By a symmetric argument, it also holds that

$$\left\| \boldsymbol{\beta}' - \hat{\boldsymbol{\beta}} \right\|_\infty \le \|\hat{\boldsymbol{\alpha}} - \boldsymbol{\alpha}\|_\infty, \tag{56}$$

where $\boldsymbol{\beta}'$ denotes the output of the Sinkhorn update for the second dual variable given the first dual variable $\boldsymbol{\alpha}$. It then follows that, for all $k \ge 0$,

$$\|\boldsymbol{\alpha}_{k+1} - \hat{\boldsymbol{\alpha}}\|_\infty \le \|\boldsymbol{\beta}_{k+1} - \hat{\boldsymbol{\beta}}\|_\infty \le \|\boldsymbol{\alpha}_k - \hat{\boldsymbol{\alpha}}\|_\infty \le \|\boldsymbol{\beta}_k - \hat{\boldsymbol{\beta}}\|_\infty. \tag{57}$$

By induction, from the above, we can deduce **(i)**. $\qquad\qquad\qquad\qquad\qquad\qquad\qquad\qquad\qquad\qquad\qquad\square$

***Proof of Theorem 2.6.*** We first claim the following: (16) holds if all Sinkhorn iterates $(\boldsymbol{\alpha}_k, \boldsymbol{\beta}_k)$ for $k \ge k_0$ as well as the potential $(\boldsymbol{\alpha}^*, \boldsymbol{\beta}^*)$ are $\varepsilon$-tame. Before proving this claim, we will first deduce parts **(i)**-**(ii)** from this claim. First we remark that there are some well-known results from the optimization literature that are directly applicable to the generalized Sinkhorn algorithm (6). Since each sub-problem in (6) has a unique solution due to strong concavity of the block-restricted dual objective, asymptotic convergence to the critical point of (6) follows from a general result for alternating maximization (e.g., Prop. 2.7.1 in (Bertsekas, 1997)). Every critical point of the dual objective is an MLE, which is a global optimum by Lemma 2.4. Thus it follows that $\boldsymbol{\alpha}_k \oplus \boldsymbol{\beta}_k \to \boldsymbol{\alpha}^* \oplus \boldsymbol{\beta}^*$ as $k \to \infty$ entrywise. In particular, if we choose $\varepsilon > 0$ small enough so that $(\mathbf{r}, \mathbf{c})$ is $\varepsilon$-tame (i.e., $\phi(A_\varepsilon) \le \boldsymbol{\alpha}^* \oplus \boldsymbol{\beta}^* \le \phi(B_\varepsilon)$), then there exists $k_0 \ge 1$ such that $\phi(A_{\varepsilon/2}) \le \boldsymbol{\alpha}_k \oplus \boldsymbol{\beta}_k \le \phi(B_{\varepsilon/2})$ for all $k \ge k_0$. Then **(i)** follows from the claim. For **(ii)**, the hypothesis of the claim is directly justified by Lemma 3.1.

It now suffices to show the claim. Our analysis for this is inspired by the proof of linear convergence of Sinkhorn algorithm for entropic optimal transport due to Carlier (Carlier, 2022). For simplicity denote $F := -g^{\mathbf{r}, \mathbf{c}}$. Consider the following centered Sinkhorn iterates $(\tilde{\boldsymbol{\alpha}}_k, \tilde{\boldsymbol{\beta}}_k)$ for $k \ge 1$ where $(\tilde{\boldsymbol{\alpha}}_0, \tilde{\boldsymbol{\beta}}_0) = (\boldsymbol{\alpha}_0, \boldsymbol{\beta}_0)$ and for $k \ge 1$, $(\tilde{\boldsymbol{\alpha}}_k, \tilde{\boldsymbol{\beta}}_k)$ is obtained from $\tilde{\boldsymbol{\beta}}_{k-1}$ by the same Sinkhorn update in (6) but follow by centering (adding and subtracting the same constants to $\tilde{\boldsymbol{\alpha}}_k$ and $\tilde{\boldsymbol{\beta}}_k$, respectively) so that $\langle \mathbf{u}, \tilde{\boldsymbol{\alpha}}_k \rangle = 0$. By an induction, it is easy to verify

$$\boldsymbol{\alpha}_k \oplus \boldsymbol{\beta}_k = \tilde{\boldsymbol{\alpha}}_k \oplus \tilde{\boldsymbol{\beta}}_k \qquad \text{for all } k \ge 0.$$

In particular, $F(\boldsymbol{\alpha}_k, \boldsymbol{\beta}_k) = F(\tilde{\boldsymbol{\alpha}}_k, \tilde{\boldsymbol{\beta}}_k)$ for all $k \ge 0$.

Denote $\sigma_\pm^2 = \sigma_\pm^2(\varepsilon)$, which are defined in the statement. Note that

$$\nabla_{\boldsymbol{\alpha}}^2 F(\boldsymbol{\alpha}, \boldsymbol{\beta}) = \text{diag}\left( \mathbf{u}(i) \sum_j \psi''(\boldsymbol{\alpha}(i) + \boldsymbol{\beta}(j)) \mathbf{C}_{ij} \mathbf{v}(j); i \right),$$

$$\nabla_{\boldsymbol{\beta}}^2 F(\boldsymbol{\alpha}, \boldsymbol{\beta}) = \text{diag}\left( \mathbf{v}(j) \sum_i \psi''(\boldsymbol{\alpha}(i) + \boldsymbol{\beta}(j)) \mathbf{C}_{ij} \mathbf{u}(i); j \right).$$

If $(\boldsymbol{\alpha}, \boldsymbol{\beta})$ and $(\boldsymbol{\alpha}', \boldsymbol{\beta}')$ are both $\varepsilon$-tame, then so is their convex combination. Hence $F(\boldsymbol{\alpha}, \cdot)$ restricted on its $j$th coordinate is $\mathbf{v}(j)\mathbf{C}_{\min}\mathbf{u}_\bullet \sigma_-^2$-strongly convex and $\mathbf{v}(j)\mathbf{C}_{\max}\mathbf{u}_\bullet \sigma_+^2$-smooth on the secant line between $(\boldsymbol{\alpha}, \boldsymbol{\beta})$ and $(\boldsymbol{\alpha}', \boldsymbol{\beta}')$. Denote

$$\tilde{\boldsymbol{\beta}}_{t;j} := (\tilde{\boldsymbol{\beta}}_t(1), \dots, \tilde{\boldsymbol{\beta}}_t(j), \tilde{\boldsymbol{\beta}}_{t+1}(j+1), \dots, \tilde{\boldsymbol{\beta}}_{t+1}(n))$$

for $j = 0, \ldots, n-1$ and let $\tilde{\boldsymbol{\beta}}_{t,n} = \tilde{\boldsymbol{\beta}}_{t+1}$. Notice that the dual problem (10) restricted to optimizing for $\boldsymbol{\beta}$ is separable over each coordinate of $\boldsymbol{\beta}$. Hence the first-order optimality of $\tilde{\boldsymbol{\beta}}_{t+1}$ implies $\nabla_{\boldsymbol{\beta}} F(\tilde{\boldsymbol{\alpha}}_t, \tilde{\boldsymbol{\beta}}_{t+1}) = \mathbf{0}$, which in turn implies

$$\nabla_{\boldsymbol{\beta}(j)} F(\boldsymbol{\alpha}, \boldsymbol{\beta}) \Big|_{\boldsymbol{\beta} = \tilde{\boldsymbol{\beta}}_{t;j}} = 0 \qquad \text{for } j = 1, \ldots, n.$$

Hence by the second-order growth property,

$$F(\boldsymbol{\alpha}_t, \tilde{\boldsymbol{\beta}}_{t;j}) - F(\boldsymbol{\alpha}_t, \tilde{\boldsymbol{\beta}}_{t;j+1}) \geq \frac{\mathbf{C}_{\min}\mathbf{u}_\bullet}{2} |\tilde{\boldsymbol{\beta}}_t(j) - \tilde{\boldsymbol{\beta}}_{t+1}(j)|^2 \mathbf{v}(j)$$

for all $j = 1, \ldots, n-1$. Summing over $j$, we obtain

$$F(\tilde{\boldsymbol{\alpha}}_t, \tilde{\boldsymbol{\beta}}_t) - F(\tilde{\boldsymbol{\alpha}}_t, \tilde{\boldsymbol{\beta}}_{t+1}) \geq \frac{\mathbf{u}_\bullet \mathbf{C}_{\min}\sigma_-^2}{2} \|\tilde{\boldsymbol{\beta}}_{t+1} - \tilde{\boldsymbol{\beta}}_t\|_{\mathbf{v}}^2,$$

where we denote the $\mathbf{v}$-weighted norm $\|\cdot\|_{\mathbf{v}}$ on $\mathbb{R}^n$ by $\|\mathbf{a}\|_{\mathbf{v}} := \sqrt{\sum_j \mathbf{a}(j)^2 \mathbf{v}(j)}$. Define the $\mathbf{u}$-weighted norm $\|\cdot\|_{\mathbf{u}}$ on $\mathbb{R}^m$ similarly. Then using a similar argument as before with $\nabla_{\boldsymbol{\alpha}} F(\tilde{\boldsymbol{\alpha}}_{t+1}, \tilde{\boldsymbol{\beta}}_{t+1}) = \mathbf{0}$, we get

$$F(\tilde{\boldsymbol{\alpha}}_t, \tilde{\boldsymbol{\beta}}_{t+1}) - F(\tilde{\boldsymbol{\alpha}}_{t+1}, \tilde{\boldsymbol{\beta}}_{t+1}) \geq \frac{\mathbf{v}_\bullet \mathbf{C}_{\min}\sigma_-^2}{2} \|\tilde{\boldsymbol{\alpha}}_{t+1} - \tilde{\boldsymbol{\alpha}}_t\|_{\mathbf{u}}^2.$$

Combining the two inequalities above and recalling $\Delta_t = F(\tilde{\boldsymbol{\alpha}}_t, \tilde{\boldsymbol{\beta}}_t) - F(\boldsymbol{\alpha}^*, \boldsymbol{\beta}^*)$, we obtain

$$\frac{\Delta_t - \Delta_{t+1}}{\mathbf{C}_{\min}\sigma_-^2} \geq \frac{\mathbf{v}_\bullet}{2} \|\tilde{\boldsymbol{\alpha}}_{t+1} - \tilde{\boldsymbol{\alpha}}_t\|_{\mathbf{u}}^2 + \frac{\mathbf{u}_\bullet}{2} \|\tilde{\boldsymbol{\beta}}_{t+1} - \tilde{\boldsymbol{\beta}}_t\|_{\mathbf{v}}^2. \tag{58}$$

Next, note that $\psi$ is $\sigma_-^2$-strongly convex and $\sigma_+^2$-smooth on $[\phi(A_\varepsilon), \phi(B_\varepsilon)]$. In particular, for each $x, y$ in that interval,

$$\psi(x) - \psi(y) \geq \psi'(y)(x - y) + \frac{\sigma_-^2}{2}(x - y)^2. \tag{59}$$

Then $\varepsilon$-tameness, (59), and $\langle \mathbf{u}, \tilde{\boldsymbol{\alpha}}_t \rangle = 0 = \langle \mathbf{u}, \boldsymbol{\alpha}^* \rangle$ give

$$\frac{1}{\mathbf{C}_{\min}} \sum_{i,j} (\psi(\boldsymbol{\alpha}^*(i) + \boldsymbol{\beta}^*(j)) - \psi(\tilde{\boldsymbol{\alpha}}_t(i) + \tilde{\boldsymbol{\beta}}_t(j))) \mathbf{W}_{ij}$$

$$\geq \sum_{i,j} \psi'(\tilde{\boldsymbol{\alpha}}_t(i) + \tilde{\boldsymbol{\beta}}_t(j))(\boldsymbol{\alpha}^*(i) + \boldsymbol{\beta}^*(j) - \tilde{\boldsymbol{\alpha}}_t(i) - \tilde{\boldsymbol{\beta}}_t(j)) \mathbf{u}(i)\mathbf{v}(j)$$

$$+ \frac{\sigma_-^2}{2} \underbrace{\|((\boldsymbol{\alpha}^* \oplus \boldsymbol{\beta}^*) - (\tilde{\boldsymbol{\alpha}}_t \oplus \tilde{\boldsymbol{\beta}}_t))\|_{\mathbf{uv}^\top}^2}_{\geq \mathbf{v}_\bullet \|\boldsymbol{\alpha}^* - \tilde{\boldsymbol{\alpha}}_t\|_{\mathbf{u}}^2 + \mathbf{u}_\bullet \|\boldsymbol{\beta}^* - \tilde{\boldsymbol{\beta}}_t\|_{\mathbf{v}}^2}. \tag{60}$$

Then using $\nabla_{\boldsymbol{\alpha}} F(\tilde{\boldsymbol{\alpha}}_t, \tilde{\boldsymbol{\beta}}_t) = \nabla_{\boldsymbol{\beta}} F(\tilde{\boldsymbol{\alpha}}_t, \tilde{\boldsymbol{\beta}}_{t+1}) = \mathbf{0}$, we can deduce the following strong-convexity-type inequality

$$\begin{aligned}
-\Delta_t &= \langle (\tilde{\boldsymbol{\alpha}}_t, \tilde{\boldsymbol{\beta}}_t) - (\boldsymbol{\alpha}^*, \boldsymbol{\beta}^*), (\mathbf{r}, \mathbf{c}) \rangle \\
&\quad + \sum_{i,j} \psi(\boldsymbol{\alpha}^*(i) + \boldsymbol{\beta}^*(j)) - \psi(\tilde{\boldsymbol{\alpha}}_t(i) + \tilde{\boldsymbol{\beta}}_t(j)) \mathbf{W}_{ij} \\
&\geq \underbrace{\langle \nabla_{\boldsymbol{\alpha}} F(\tilde{\boldsymbol{\alpha}}_t, \tilde{\boldsymbol{\beta}}_t), \boldsymbol{\alpha}^* - \tilde{\boldsymbol{\alpha}}_t \rangle}_{=\mathbf{0}} + \langle \nabla_{\boldsymbol{\beta}} F(\tilde{\boldsymbol{\alpha}}_t, \tilde{\boldsymbol{\beta}}_t), \boldsymbol{\beta}^* - \tilde{\boldsymbol{\beta}}_t \rangle \\
&\quad + \frac{\mathbf{C}_{\min}\sigma_-^2}{2} \left[ \mathbf{v}_\bullet \|\boldsymbol{\alpha}^* - \tilde{\boldsymbol{\alpha}}_t\|_{\mathbf{u}}^2 + \mathbf{u}_\bullet \|\boldsymbol{\beta}^* - \tilde{\boldsymbol{\beta}}_t\|_{\mathbf{v}}^2 \right] \\
&\overset{(a)}{\geq} -\frac{1}{2\mathbf{u}_\bullet \mathbf{C}_{\min}\sigma_-^2} \|\nabla_{\boldsymbol{\beta}} F(\tilde{\boldsymbol{\alpha}}_t, \tilde{\boldsymbol{\beta}}_t) - \nabla_{\boldsymbol{\beta}} F(\tilde{\boldsymbol{\alpha}}_t, \tilde{\boldsymbol{\beta}}_{t+1})\|_{1/\mathbf{v}}^2 \\
&\overset{(b)}{\geq} -\frac{\mathbf{C}_{\max}^2 \sigma_+^4}{2\mathbf{C}_{\min}\sigma_-^2} \left( \mathbf{v}_\bullet \|\tilde{\boldsymbol{\alpha}}_{t+1} - \tilde{\boldsymbol{\alpha}}_t\|_{\mathbf{u}}^2 + \mathbf{u}_\bullet \|\tilde{\boldsymbol{\beta}}_{t+1} - \tilde{\boldsymbol{\beta}}_t\|_{\mathbf{v}}^2 \right)
\end{aligned}$$

where (a) follows from coordinate-wise Young's inequality ($ab \le \frac{\lambda a^2}{2} + \frac{b^2}{2\lambda}$ for $a, b \ge 0$ and $\lambda = \mathbf{u}_\bullet \mathbf{v}(j)\mathbf{C}_{\min}\sigma_-^2$ for $j$th coordinate) and (b) follows from the Lipschitz continuity of $\nabla_{\boldsymbol{\beta}} F$ and including an additional nonpositive term for the lower bound. Combining with (58) and recalling $\kappa = \frac{\mathbf{C}_{\max}}{\mathbf{C}_{\min}}$,

$$\Delta_t \le \kappa^2 (\sigma_+/\sigma_-)^4 (\Delta_t - \Delta_{t+1}).$$

Rearranging, this is $\Delta_{t+1} \le \left(1 - \kappa^{-2}(\sigma_-/\sigma_+)^4\right)\Delta_t$. A similar argument as above also shows

$$\Delta_t \ge \frac{\mathbf{C}_{\min}\sigma_-^2}{2}\|(\boldsymbol{\alpha}^* \oplus \boldsymbol{\beta}^*) - (\boldsymbol{\alpha}_t \oplus \boldsymbol{\beta}_t)\|_{\mathbf{uv}^\top}^2.$$

Combining, we obtain (16) as claimed. $\qquad\square$

***Proof of Theorem*** 2.8. By the hypothesis, $(A, B) = (0, \infty)$, $s \le \mathbf{r}(i)/\mathbf{W}_{i\bullet} \le t$ and $s \le \mathbf{c}(j)/\mathbf{W}_{\bullet j} \le t$ for all $i, j$. By Lemma 2.4, there exists a potential $(\boldsymbol{\alpha}, \boldsymbol{\beta})$ for $(\mathbf{r}, \mathbf{c})$ w.r.t. $\mathbf{W}$. Note that $\phi(z_{ij}) = \boldsymbol{\alpha}(i) + \boldsymbol{\beta}(j) \in (\phi(A), \phi(B))$ for all $i, j$.

Denote $\kappa = \mathbf{C}_{\max}/\mathbf{C}_{\min}$ and define

$$c(\lambda; s, t) := \phi\left(\frac{\kappa^2 t}{\lambda s}(t - (1 - \lambda)s)\right) - \phi((1 - \lambda)t), \tag{61}$$

which is well-defined whenever $\lambda \in \text{dom}(c(\cdot; s, t)) = \left(\frac{t(t-s)}{s(B-t)}, \frac{t-A}{t}\right) = (0, 1)$ under the hypothesis. Note that $\frac{t}{\lambda s}(t - (1 - \lambda)s) \ge t$ so we get $c(\lambda; s, t) \ge \phi(t) - \phi((1 - \lambda)t) \ge 0$. We claim that for all $i, j$ and $\lambda \in \text{dom}(c(\cdot; s, t))$,

$$\phi(s) - c(\lambda; s, t) \le \boldsymbol{\alpha}(i) + \boldsymbol{\beta}(j) \le \phi(t) + c(\lambda; s, t). \tag{62}$$

This will directly imply (23). For the second part of the conclusion, note that, since $\phi$ is continuous, we may choose $\lambda \in (0, 1)$ such that $\phi((1 - \lambda)s), \phi((1 + \lambda)t) \in \Theta^\circ$ and $\bar{c}(\lambda; t) := \phi((1 + \lambda)t) - \phi((1 - \lambda)t) \ge 0$ is so small that

$$\phi(s) - \bar{c}(\lambda; t) \in \Theta^\circ. \tag{63}$$

Also, since $0 \le c(\lambda; s, t) \le \bar{c}(\lambda; t)$, it follows that from (63), we have

$$\boldsymbol{\alpha}(i) + \boldsymbol{\beta}(j) \in [\phi(s) - \bar{c}(\lambda; t), \phi(t) + c(\lambda; s, t)] \subseteq (0, \infty) \subseteq \Theta^\circ \qquad \text{for all } i, j. \tag{64}$$

Therefore, $(\mathbf{r}, \mathbf{c})$ is $\delta$-tame for $\delta > 0$ small enough so that $A_\delta \le \phi(s) - \bar{c}(\lambda; t)$ and $\phi(t) + c(\lambda; s, t) \le B_\delta$.

Now it remains to show (62). We may assume that $0 = \boldsymbol{\alpha}(1) \le \cdots \le \boldsymbol{\alpha}(m)$ and $\boldsymbol{\beta}(1) \le \cdots \le \boldsymbol{\beta}(n)$ by shifting the potentials and permuting the rows and columns if necessary. We first note that $\boldsymbol{\beta}(n) \le \phi(t)$, which follows from

$$t\mathbf{W}_{\bullet n} \ge \mathbf{c}(n) = \sum_{i=1}^m \mathbf{W}_{in}\psi'(\boldsymbol{\alpha}(i) + \boldsymbol{\beta}(n)) \ge \sum_{i=1}^m \mathbf{W}_{in}\psi'(\boldsymbol{\beta}(n)) = \psi'(\boldsymbol{\beta}(n))\mathbf{W}_{\bullet n}. \tag{65}$$

and $\phi(t) \in \Theta^\circ$. Define

$$M := \sum_{j: \boldsymbol{\beta}(j) \ge \phi((1-\lambda)s)} \mathbf{W}_{1j}. \tag{66}$$

Note that

$$s\mathbf{W}_{1\bullet} \le \mathbf{r}(1) = \sum_{\boldsymbol{\beta}(j) < \phi((1-\lambda)s)} \mathbf{W}_{1j}\psi'(\boldsymbol{\beta}(j)) + \sum_{\boldsymbol{\beta}(j) \ge \phi((1-\lambda)s)} \mathbf{W}_{1j}\psi'(\boldsymbol{\beta}(j)) \tag{67}$$

$$\le M(t - (1 - \lambda)s) + \mathbf{W}_{1\bullet}(1 - \lambda)s. \tag{68}$$

Rearranging terms and since $\mathbf{W}_{1\bullet}, s > 0$, we deduce

$$\frac{M}{\mathbf{W}_{1\bullet}} \ge \frac{\lambda s}{t - (1 - \lambda)s} > 0. \tag{69}$$

Hence there exists an index $j^*$ such that $\boldsymbol{\beta}(j^*) \geq \phi((1 - \lambda)s)$. Let such $j^*$ be as small as possible.

Next, we claim that

$$\kappa^2 \frac{t}{\lambda s}(t - (1 - \lambda)s) \geq \psi'(\boldsymbol{\alpha}(m) + \boldsymbol{\beta}(j^*)), \tag{70}$$

To show this, observe that $\psi'(\boldsymbol{\alpha}(m) + \boldsymbol{\beta}(1)) \geq 0$ since

$$0 < s\mathbf{W}_{1\bullet} \leq \mathbf{c}(1) = \sum_{i=1}^{m} \mathbf{W}_{i1}\psi'(\boldsymbol{\alpha}(i) + \boldsymbol{\beta}(1)) \leq \psi'(\boldsymbol{\alpha}(m) + \boldsymbol{\beta}(1))\mathbf{W}_{\bullet 1}. \tag{71}$$

Hence,

$$t \geq \frac{\mathbf{r}(m)}{\mathbf{W}_{m\bullet}} \geq \sum_{j \geq j^*} \frac{\mathbf{W}_{mj}}{\mathbf{W}_{m\bullet}}\psi'(\boldsymbol{\alpha}(m) + \boldsymbol{\beta}(j^*))) + \sum_{j < j^*} \frac{\mathbf{W}_{mj}}{\mathbf{W}_{m\bullet}}\psi'(\boldsymbol{\alpha}(m) + \boldsymbol{\beta}(1)) \tag{72}$$

$$\geq \frac{\sum_{j \geq j^*} \mathbf{W}_{mj}}{\mathbf{W}_{m\bullet}}\psi'(\boldsymbol{\alpha}(m) + \boldsymbol{\beta}(j^*)). \tag{73}$$

Now from the hypothesis that $\mathbf{W} = \mathbf{C} \odot \mathbf{u}\mathbf{v}^\top$ and recalling $\kappa = \mathbf{C}_{\max}/\mathbf{C}_{\min}$,

$$\frac{\sum_{j \geq j^*} \mathbf{W}_{mj}}{\mathbf{W}_{m\bullet}} \Big/ \frac{\sum_{j \geq j^*} \mathbf{W}_{1j}}{\mathbf{W}_{1\bullet}} = \frac{\sum_{j \geq j^*} \mathbf{C}_{mj}\mathbf{v}(j)}{\sum_j \mathbf{C}_{mj}\mathbf{v}(j)} \Big/ \frac{\sum_{j \geq j^*} \mathbf{C}_{1j}\mathbf{v}(j)}{\sum_j \mathbf{C}_{1j}\mathbf{v}(j)} \geq \frac{\sum_{j \geq j^*} \mathbf{C}_{\min}\mathbf{v}(j)}{\sum_j \mathbf{C}_{\max}\mathbf{v}(j)} \Big/ \frac{\sum_{j \geq j^*} \mathbf{C}_{\max}\mathbf{v}(j)}{\sum_j \mathbf{C}_{\min}\mathbf{v}(j)} = \kappa^{-2}. \tag{74}$$

Combining with the previous inequality and using (69), we get

$$t \geq \kappa^{-2} \frac{\lambda s}{t - (1 - \lambda)s}\psi'(\boldsymbol{\alpha}(m) + \boldsymbol{\beta}(j^*)). \tag{75}$$

Simplifying the above yields (70).

Then applying $\phi$ on both sides of (70) and using $\boldsymbol{\beta}(j^*) \geq \phi((1 - \lambda)t)$, we get

$$\boldsymbol{\alpha}(m) \leq \phi\left(\frac{\kappa^2 t(t - (1 - \lambda)s)}{\lambda s}\right) - \boldsymbol{\beta}(j^*) \leq c(\lambda; s, t). \tag{76}$$

In turn, we can deduce a lower bound on $\boldsymbol{\beta}(1)$ from

$$s \leq \frac{\mathbf{c}(1)}{\mathbf{W}_{\bullet 1}} = \sum_{i=1}^{m} \frac{\mathbf{W}_{i1}}{\mathbf{W}_{\bullet 1}}\psi'(\boldsymbol{\alpha}(i) + \boldsymbol{\beta}(1)) \leq \psi'(\boldsymbol{\alpha}(m) + \boldsymbol{\beta}(1)), \tag{77}$$

which reads $\boldsymbol{\beta}(1) \geq \phi(s) - \boldsymbol{\alpha}(m) \geq \phi(s) - c(\lambda; s, t)$. Therefore, $\boldsymbol{\alpha}(1) + \boldsymbol{\beta}(1) \geq \phi(s) - c(\lambda; s, t)$ and $\boldsymbol{\alpha}(m) + \boldsymbol{\beta}(n) \leq \phi(t) + c(\lambda; s, t)$. This shows (62), as desired. □

***Proof of Theorem 2.7.*** Without loss of generality, assume $m \geq n$. Let $\mathbf{Z}$ be the SSB for margin $(\mathbf{r}, \mathbf{c})$ w.r.t. $\mathbf{W}$ and $(\boldsymbol{\alpha}, \boldsymbol{\beta})$ be a potential for $(\mathbf{r}, \mathbf{c})$ w.r.t. $\mathbf{W}$. respectively. By Lemma 2.4, we have $\mathbf{Z} = \psi'(\boldsymbol{\alpha} \oplus \boldsymbol{\beta})$. Write $\boldsymbol{\alpha} = (\alpha_1, \ldots, \alpha_m)$ and $\boldsymbol{\beta} = (\beta_1, \ldots, \beta_n)$. Without loss of generality, we assume $\boldsymbol{\alpha}_1 \leq \cdots \leq \boldsymbol{\alpha}_m$ as well as $\boldsymbol{\beta}_1 \leq \cdots \leq \boldsymbol{\beta}_n$.

Recall that for any $\lambda \in \mathbb{R}$, $(\boldsymbol{\alpha} + \lambda\mathbf{1}_m, \boldsymbol{\beta} - \lambda\mathbf{1}_n)$ is also a potential for margin $(\mathbf{r}, \mathbf{c})$ w.r.t. $\mathbf{W}$. Define functions

$$\rho(\lambda) := \frac{1}{\mathbf{u}_\bullet}\sum_i \mathbf{u}(i)\mathbf{1}(\alpha_i + \lambda \geq 0) \quad \text{and} \quad \gamma(\lambda) := \frac{1}{\mathbf{v}_\bullet}\sum_j \mathbf{v}(j)\mathbf{1}(\beta_j - \lambda \geq 0), \tag{78}$$

where $\mathbf{a}_\bullet := \langle \mathbf{1}, \mathbf{a} \rangle$ for a vector $\mathbf{a}$. Note that $\rho(\lambda)$ is a right-continuous non-decreasing stepfunction and $\gamma(\lambda)$ is a left-continuous non-increasing stepfunction. If $\lambda < \min(\beta_1, -\alpha_m)$, then $\alpha_m + \lambda < 0$ and $\beta_1 - \lambda > 0$, so $\rho(\lambda) = 0$ and $\gamma(\lambda) = 1$. Similarly, if $\lambda > \max(-\alpha_1, \beta_n)$, then $\alpha_1 + \lambda > 0$ and $\beta_n - \lambda < 0$, so $\gamma(\lambda) = 1$ and $\gamma(\lambda) = 0$. Hence there must be an intermediate value of $\lambda$, say $\lambda^*$, where the graphs of $\rho(\lambda)$ and $\gamma(\lambda)$ cross.

Then near $\lambda^*$, either (1) $\rho$ is constant and $\gamma$ drops down vertically or (2) $\gamma$ is constant and $\rho$ jumps up vertically. Without loss of generality, we will assume (1) holds (the other case can be handled by a symmetric argument). Then we have

$$\rho = w \frac{1}{\mathbf{v}_\bullet} \sum_j \mathbf{v}(j) \mathbf{1}(\beta_j^* = 0) + \frac{1}{\mathbf{v}_\bullet} \sum_j \mathbf{v}(j) \mathbf{1}(\beta_j^* > 0) \quad \text{for some } w \in [0,1]. \tag{79}$$

In the remainder of the proof, we will consider the shifted potential $(\boldsymbol{\alpha}^*, \boldsymbol{\beta}^*) := (\boldsymbol{\alpha} + \lambda^* \mathbf{1}_m, \boldsymbol{\beta} - \lambda^* \mathbf{1}_n)$. We will denote $\boldsymbol{\alpha}^* = (\alpha_1^*, \dots, \alpha_m^*)$ and $\boldsymbol{\beta}^* = (\beta_1^*, \dots, \beta_n^*)$ and $\delta := \rho(\lambda^*)$ and $\gamma := \gamma(\lambda^*)$.

Fix $\varepsilon > 0$ sufficiently small. Note that $\phi(\varepsilon) \to -\infty$ as $\varepsilon \searrow 0$ and $\phi(B - \varepsilon) \to \infty$ as $\varepsilon \searrow 0$. Hence there are constants $c_1, c_2$ (depending on $\varepsilon$) such that

$$\phi(c_1 \varepsilon) + \phi(B - \varepsilon) \le \phi(s - \varepsilon), \tag{80}$$
$$\phi(\varepsilon) + \phi(B - c_2 \varepsilon) \ge \phi(t + \varepsilon). \tag{81}$$

Then, for fixed $\varepsilon, c_1, c_2$, we may choose $c_3, c_4$ small enough so that

$$\phi(c_3 \varepsilon) + \phi(B - c_2 \varepsilon) \le \phi(s - \varepsilon), \tag{82}$$
$$\phi(c_1 \varepsilon) + \phi(B - c_4 \varepsilon) \ge \phi(t + \varepsilon). \tag{83}$$

We claim that

$$\phi(c_1 \varepsilon) \overset{(a)}{\le} \alpha_1^* \le \alpha_m^* \overset{(b)}{\le} \phi(B - c_2 \varepsilon), \tag{84}$$
$$\phi(c_3 \varepsilon) \overset{(c)}{\le} \beta_1^* \le \beta_n^* \overset{(d)}{\le} \phi(B - c_4 \varepsilon). \tag{85}$$

Since $\phi(\mathbf{Z}_{ij}) = \alpha_i^* + \beta_j^*$ and $\phi$ is non-decreasing, the assertion follows immediately from this claim. Inequalities (c) and (d) follow easily from assuming (a) and (b). Indeed, suppose (d) is not true while (a) holds. Then

$$t \mathbf{W}_{\bullet n} \ge \mathbf{c}(n) = \sum_i \mathbf{W}_{in} \psi'(\alpha_i^* + \beta_n^*) \ge \mathbf{W}_{\bullet n} \psi'\big(\phi(c_1 \varepsilon) + \phi(B - c_4 \varepsilon)\big) \ge (t + \varepsilon) \mathbf{W}_{\bullet n}, \tag{86}$$

which is a contradiction. Thus if we show (a) holds, then (d) also holds. Similarly, if (c) is not true while (b) holds, then

$$s \mathbf{W}_{\bullet 1} \le \mathbf{c}(1) = \sum_i \mathbf{W}_{i1} \psi'(\alpha_i^* + \beta_1^*) \le \mathbf{W}_{\bullet 1} \psi'\big(\phi(B - c_2 \varepsilon) + \phi(c_3 \varepsilon)\big) \le (s - \varepsilon) \mathbf{W}_{\bullet 1}, \tag{87}$$

which is a contradiction. Therefore, it remains to show (84)(a)-(b).

Suppose for contradiction (a) does not hold, i.e., $\alpha_1^* < \phi(c_1 \varepsilon)$. Then necessarily $\beta_n^* > \phi(B - \varepsilon)$, since otherwise for all $1 \le j \le n$,

$$z_{1,j} = \psi'(\alpha_1^* + \beta_j^*) \le \psi'(\alpha_1^* + \beta_n^*) \le \psi'\big(\phi(c_1 \varepsilon) + \phi(B - \varepsilon)\big) \le \psi'(\phi(s - \varepsilon)) = s - \varepsilon. \tag{88}$$

This implies $s \mathbf{W}_{1 \bullet} \le \mathbf{r}(1) \le (s - \varepsilon) \mathbf{W}_{1 \bullet}$, a contradiction.

Now since $\alpha_1^* < \phi(c_1 \varepsilon)$ and $\beta_n^* > \phi(B - \varepsilon)$,

$$\mathbf{Z}_{i,n} = \psi'(\alpha_i^* + \beta_n^*) \ge \psi'(\beta_n^*) = \psi'(\phi(B - \varepsilon)) = B - \varepsilon \quad \text{if } \alpha_i^* \ge 0, \tag{89}$$
$$\mathbf{Z}_{1,j} = \psi'(\alpha_1^* + \beta_j^*) \le \psi'(\alpha_1^*) = \psi'(\phi(c_1 \varepsilon)) = c_1 \varepsilon \quad \text{if } \beta_j^* \le 0. \tag{90}$$

Then we have

$$t \mathbf{C}_{\max} \mathbf{v}(n) \mathbf{u}_\bullet \ge t \mathbf{W}_{\bullet n} \ge \mathbf{c}(n) = \sum_{i; \alpha_i \ge 0} \mathbf{W}_{i,n} \mathbf{Z}_{i,n} + \sum_{i; \alpha_i^* < 0} \mathbf{W}_{i,n} \mathbf{Z}_{i,n} \tag{91}$$

$$\ge (B - \varepsilon) \sum_{i; \alpha_i^* \ge 0} \mathbf{W}_{i,n} + \mathbf{Z}_{1,n} \sum_{i; \alpha_i^* < 0} \mathbf{W}_{i,n}, \tag{92}$$

$$\ge (B - \varepsilon) \mathbf{v}(n) \sum_{i; \alpha_i^* \ge 0} \mathbf{C}_{i,n} \mathbf{u}(i) + \mathbf{Z}_{1,n} \mathbf{v}(n) \sum_{i; \alpha_i^* < 0} \mathbf{C}_{i,n} \mathbf{u}(i), \tag{93}$$

$$\ge (B - \varepsilon) \mathbf{v}(n) \mathbf{C}_{\min} \sum_{i; \alpha_i^* \ge 0} \mathbf{u}(i) + \mathbf{Z}_{1,n} \mathbf{v}(n) \mathbf{C}_{\min} \sum_{i; \alpha_i^* < 0} \mathbf{u}(i), \tag{94}$$

where for the last inequality, we used the fact that $\mathbf{Z}_{i,j} \in \text{dom}(f') \subseteq [0, \infty)$. This gives

$$t \geq (B - \varepsilon) \frac{\mathbf{C}_{\min}}{\mathbf{C}_{\max}} \rho + \mathbf{Z}_{1,n} \frac{\mathbf{C}_{\min}}{\mathbf{C}_{\max}} (1 - \rho). \tag{95}$$

Similarly but now using (79),

$$s\mathbf{C}_{\min}\mathbf{u}(1)\mathbf{v}_\bullet \leq s\mathbf{W}_{1\bullet} \tag{96}$$

$$\leq \mathbf{r}(1) = \left( \sum_{j;\beta_j < 0} \mathbf{W}_{1,j}\mathbf{Z}_{1,j} + (1 - w) \sum_{j;\beta_j^* = 0} \mathbf{W}_{1,j}\mathbf{Z}_{1,j} \right) + \left( w \sum_{j;\beta_j^* = 0} \mathbf{W}_{1,j}\mathbf{Z}_{1,j} + \sum_{j;\beta_j^* > 0} \mathbf{W}_{1,j}\mathbf{Z}_{1,j} \right) \tag{97}$$

$$\leq c_1\mathbf{u}(1)\mathbf{C}_{\max}\left( \sum_{j;\beta_j^* < 0} \mathbf{v}(j) + (1 - w) \sum_{j;\beta_j^* = 0} \mathbf{v}(j) \right) + \mathbf{Z}_{1,n}\mathbf{u}(1)\mathbf{C}_{\max}\left( w \sum_{j;\beta_j^* = 0} \mathbf{v}(j) + \sum_{j;\beta_j^* > 0} \mathbf{v}(j) \right) \tag{98}$$

$$\leq c_1\mathbf{u}(1)\mathbf{C}_{\max}(1 - \rho)\mathbf{v}_\bullet + \mathbf{Z}_{1,n}\mathbf{u}(1)\mathbf{C}_{\max}\rho\mathbf{v}_\bullet. \tag{99}$$

Simplifying, we get

$$s \leq c_1 \frac{\mathbf{C}_{\max}}{\mathbf{C}_{\min}}(1 - \rho) + \mathbf{Z}_{1,n} \frac{\mathbf{C}_{\max}}{\mathbf{C}_{\min}} \rho. \tag{100}$$

Thus far we have obtained the following two inequalities

$$\tilde{t} := \kappa t \geq \rho(B - \varepsilon) + (1 - \rho)\mathbf{Z}_{1,n}, \qquad \tilde{s} := \kappa^{-1}s \leq (1 - \rho)c_1 + \rho\mathbf{Z}_{1,n}. \tag{101}$$

Denote $\tau = \mathbf{Z}_{1,n}$. Since $\rho \in [0, 1]$,

$$\tilde{t} + \varepsilon \geq \rho B + (1 - \rho)\tau, \qquad \tilde{s} - c_1 \leq \rho\tau. \tag{102}$$

This yields

$$\tilde{t} + \varepsilon \geq \rho B + (1 - \rho)\tau \geq 2\sqrt{B\rho\tau} - \rho\tau \geq 2\sqrt{B(\tilde{s} - c_1)} - (\tilde{s} - c_1), \tag{103}$$

where the first and the last inequalities above is from (102). The middle inequality above uses $\frac{\rho B + \tau}{2} \geq \sqrt{\rho B \tau}$ and the fact that the function $x \mapsto 2\sqrt{Bx} - x$ is increasing for $x \in [0, B]$. Thus if

$$\tilde{t} + \varepsilon < 2\sqrt{B(\tilde{s} - c_1)} - (\tilde{s} - c_1), \tag{104}$$

then this leads to a contradiction. Note that (104) holds for $\varepsilon, c_1$ sufficiently small if $\tilde{t} < 2\sqrt{B\tilde{s}} - \tilde{s}$, or $(\tilde{s} + \tilde{t})^2 < 4B\tilde{s}$. The last condition reads

$$(\kappa^{-1}s + \kappa t)^2 < 4B\kappa^{-1}s \tag{105}$$

Thus we conclude that (84) (a) hold. An entirely similar argument also shows (84) (b). This completes the proof. $\square$

## C. Additional Numerical Experiments

In this section, we consider a total of twelve settings (two of which we already cover earlier in Section 4) in dimension 100, combining the three hyperparameter settings:

- **Marginal distributions**: (1) Each coordinate is uniformly sampled from (0,1) and then normalized; and (2) Gaussian distributions with 1- or 2-modes.

- **Cost functions**: (1) Each coordinate is uniformly sampled from (0,1); (2) $L^2$-cost function

- **Entropic regularization**: (1) $\varepsilon = 1$, (2) $\varepsilon = 10$, and (3) $\varepsilon = 100$.

As before, we consider the following six strictly convex f-divergences:

- KL divergence: $f(x) = x \log x$,

- Jeffreys divergence: $f(x) = (x-1) \log x$,

- Jensen-Shannon divergence:
  $f(x) = \frac{1}{2}(x \log x - (x+1) \log(\frac{x+1}{2}))$,

- Neyman $\chi^2$ divergence: $f(x) = (x-1)^2$,

- Binomial divergence: $f(x) = 10 \log(\frac{1+e^x}{2})$, and

- Squared Hellinger divergence: $f(x) = \frac{1}{2}(\sqrt{x} - 1)^2$.

Each experiment is organized as a (3,3) plot. Marginal distributions are shown in the subplot (1,1), the cost matrix $\mathbf{C}$ is shown at (1,2), and the $L^1$ gradient norm of the dual objective (also the $L^1$ marginal error) is shown at (1,3). The second and the third rows show the optimal transport maps found by the GSA with the corresponding divergence functions. Figures 4 through 15 cover all the remaining ten of the twelve combinations.

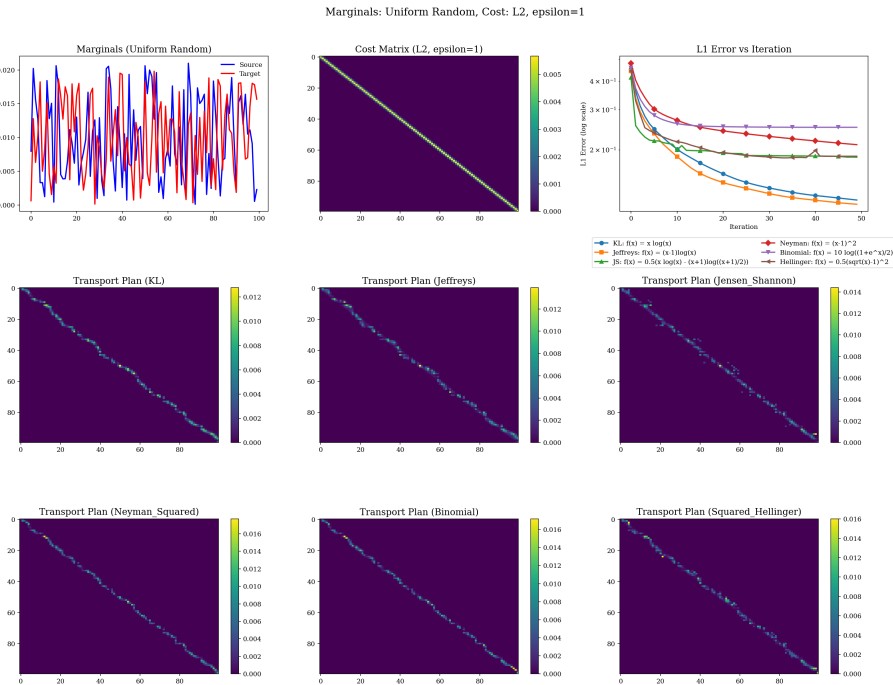

*Figure 4.* Convergence of dual objective gradient norms of GSSB, transportation maps for various f-divergences for uniformly random marginals in dimensions $m = 100$ and $n = 100$, $L^2$ cost function, and entropic regularization parameter $\epsilon = 1$. The marginals are shown in the subplot (1,1), the cost matrix $\mathbf{C}$ is shown at (1,2), and the $L^1$ gradient norm of the dual objective (also the $L^1$ marginal error) is shown at (1,3). The second and third rows show the optimal transport maps found by GSA with the corresponding divergence functions.

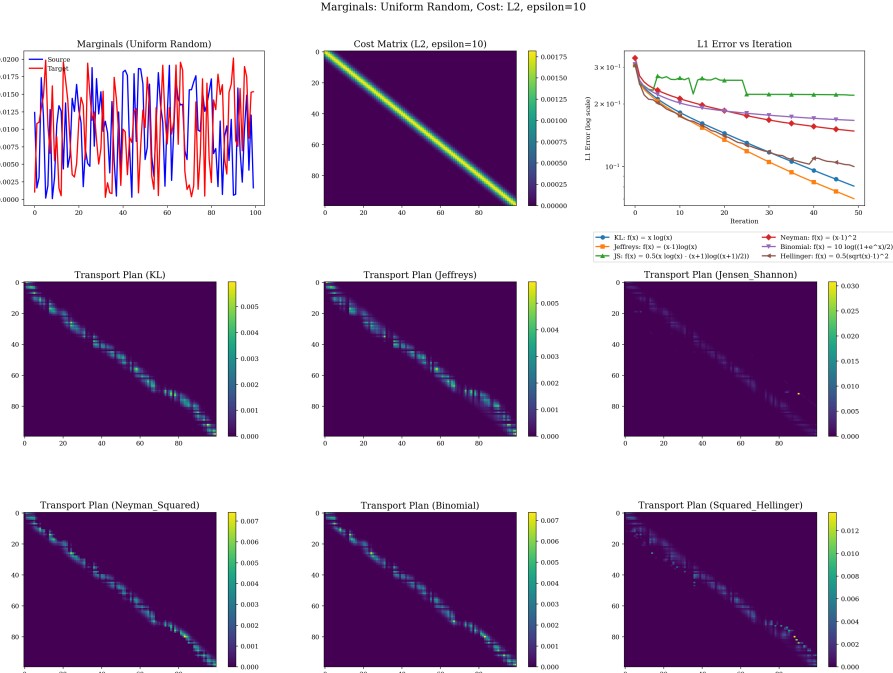

*Figure 5.* Convergence of dual objective gradient norms of GSSB, transportation maps for various f-divergences for uniformly random marginals in dimensions $m = 100$ and $n = 100$, $L^2$ cost function, and entropic regularization parameter $\epsilon = 10$. The marginals are shown in the subplot (1,1), the cost matrix **C** is shown at (1,2), and the $L^1$ gradient norm of the dual objective (also the $L^1$ marginal error) is shown at (1,3). The second and third rows show the optimal transport maps found by GSA with the corresponding divergence functions.

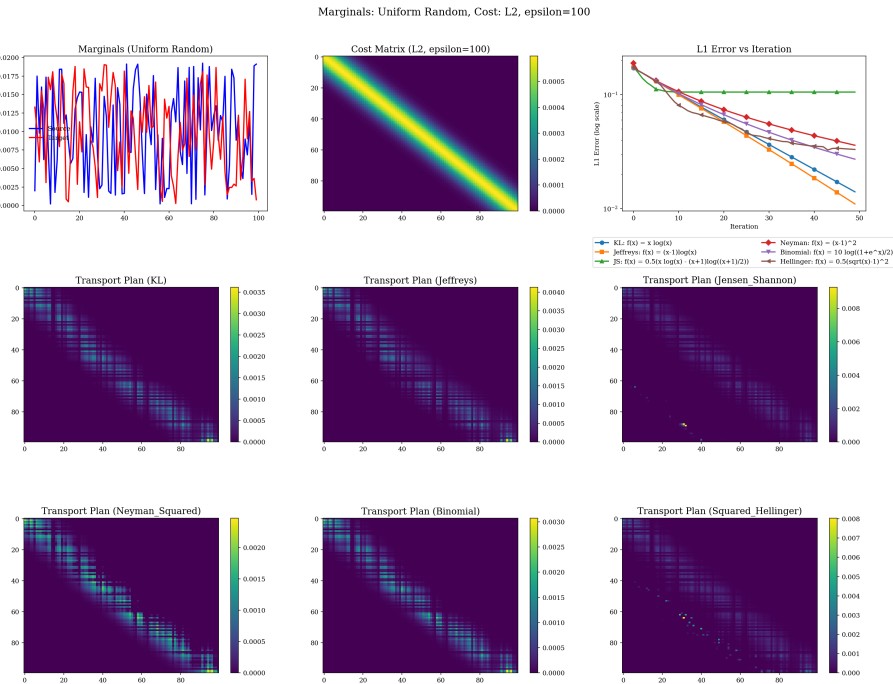

*Figure 6.* Convergence of dual objective gradient norms of GSSB, transportation maps for various f-divergences for uniformly random marginals in dimensions $m = 100$ and $n = 100$, $L^2$ cost function, and entropic regularization parameter $\epsilon = 100$. The marginals are shown in the subplot (1,1), the cost matrix **C** is shown at (1,2), and the $L^1$ gradient norm of the dual objective (also the $L^1$ marginal error) is shown at (1,3). The second and third rows show the optimal transport maps found by GSA with the corresponding divergence functions.

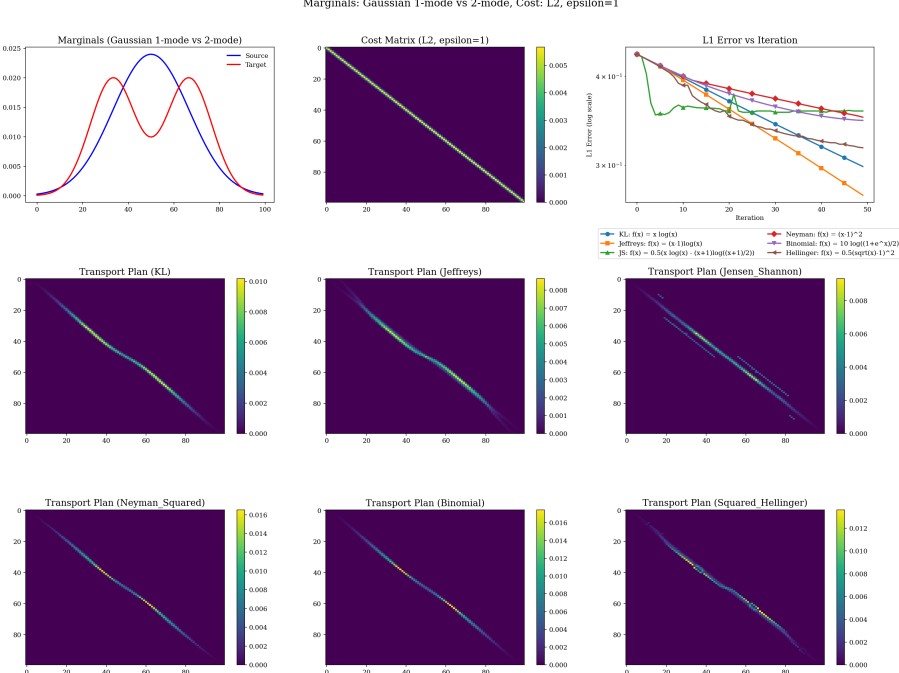

*Figure 7.* Convergence of dual objective gradient norms of GSSB, transportation maps for various f-divergences for Gaussian marginals in dimensions $m = 100$ and $n = 100$, $L^2$ cost function, and entropic regularization parameter $\epsilon = 1$. The marginals are shown in the subplot (1,1), the cost matrix **C** is shown at (1,2), and the $L^1$ gradient norm of the dual objective (also the $L^1$ marginal error) is shown at (1,3). The second and third rows show the optimal transport maps found by GSA with the corresponding divergence functions.

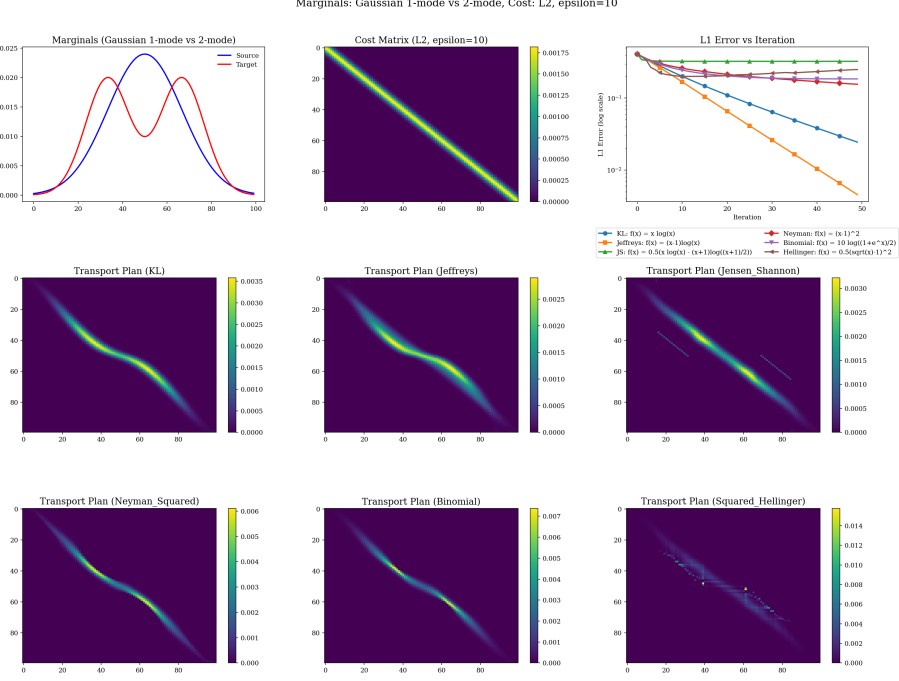

*Figure 8.* Convergence of dual objective gradient norms of GSSB, transportation maps for various f-divergences for Gaussian marginals in dimensions $m = 100$ and $n = 100$, $L^2$ cost function, and entropic regularization parameter $\epsilon = 10$. The marginals are shown in the subplot (1,1), the cost matrix **C** is shown at (1,2), and the $L^1$ gradient norm of the dual objective (also the $L^1$ marginal error) is shown at (1,3). The second and third rows show the optimal transport maps found by GSA with the corresponding divergence functions.

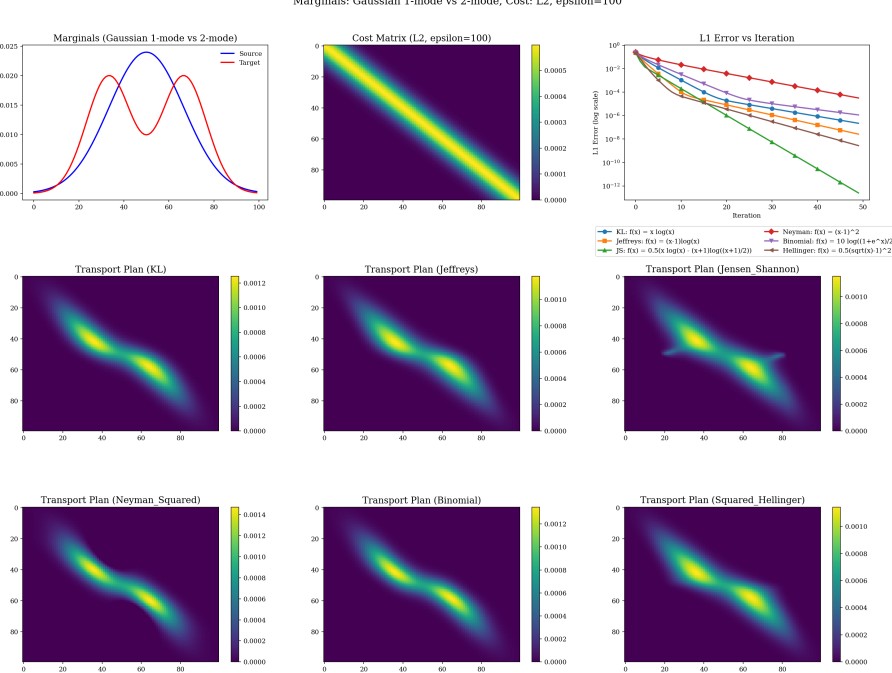

*Figure 9.* Convergence of dual objective gradient norms of GSSB, transportation maps for various f-divergences for Gaussian marginals in dimensions $m = 100$ and $n = 100$, $L^2$ cost function, and entropic regularization parameter $\epsilon = 100$. The marginals are shown in the subplot (1,1), the cost matrix **C** is shown at (1,2), and the $L^1$ gradient norm of the dual objective (also the $L^1$ marginal error) is shown at (1,3). The second and third rows show the optimal transport maps found by GSA with the corresponding divergence functions.

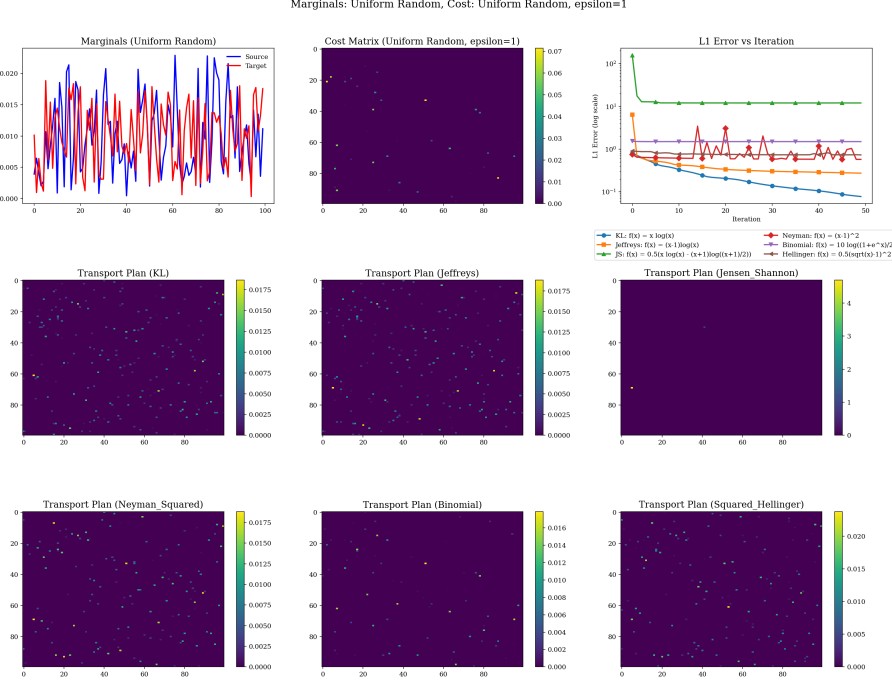

*Figure 10.* Convergence of dual objective gradient norms of GSSB, transportation maps for various f-divergences for uniformly random marginals in dimensions $m = 100$ and $n = 100$, uniformly random cost function, and entropic regularization parameter $\epsilon = 1$. The marginals are shown in the subplot (1,1), the cost matrix **C** is shown at (1,2), and the $L^1$ gradient norm of the dual objective (also the $L^1$ marginal error) is shown at (1,3). The second and third rows show the optimal transport maps found by GSA with the corresponding divergence functions.

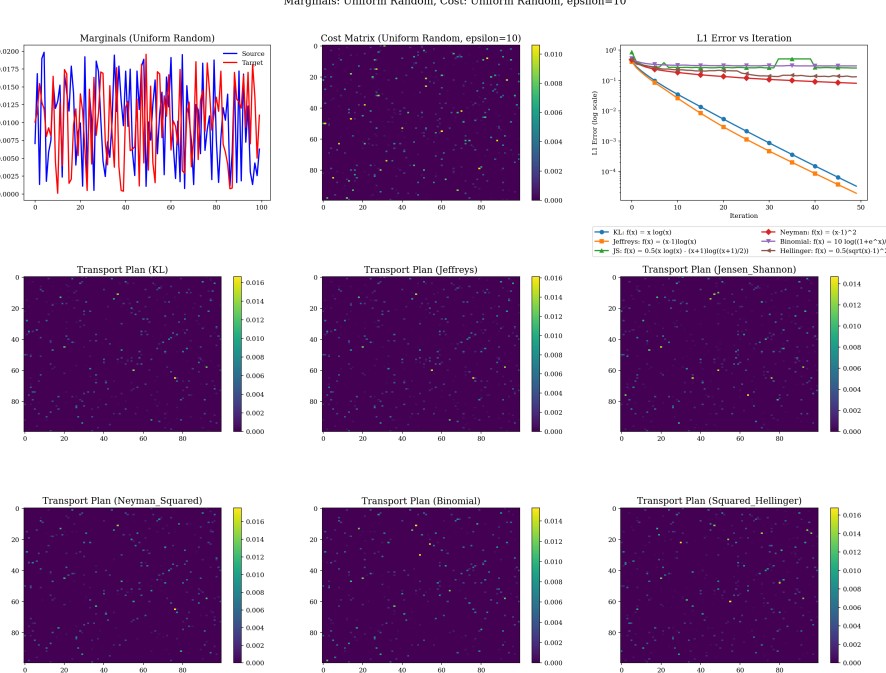

*Figure 11.* Convergence of dual objective gradient norms of GSSB, transportation maps for various f-divergences for uniformly random marginals in dimensions $m = 100$ and $n = 100$, uniformly random cost function, and entropic regularization parameter $\epsilon = 10$. The marginals are shown in the subplot (1,1), the cost matrix **C** is shown at (1,2), and the $L^1$ gradient norm of the dual objective is shown at (1,3). The second and third rows show the optimal transport maps found by GSA with the corresponding divergence functions.

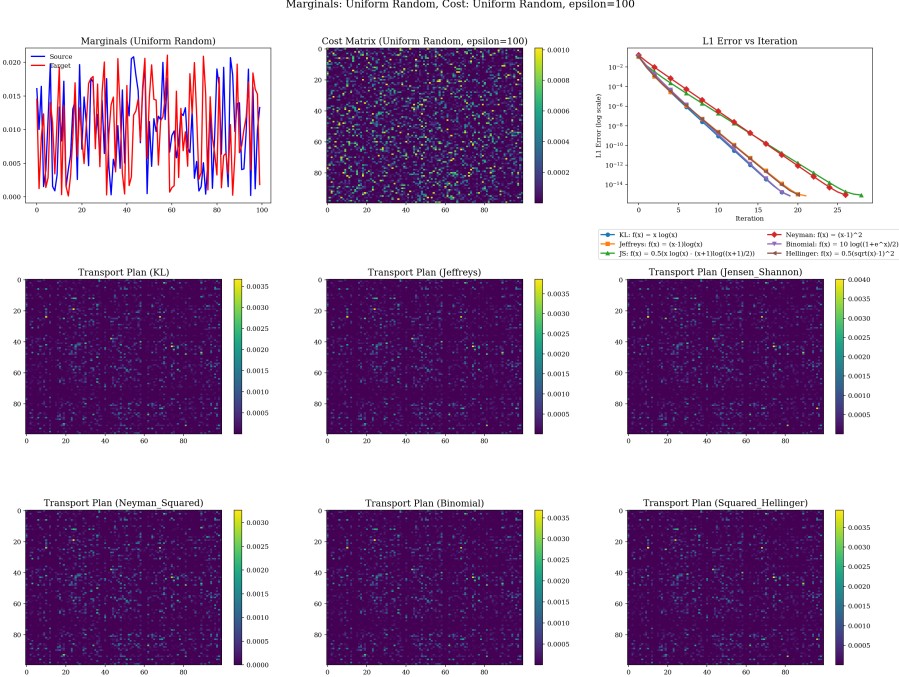

*Figure 12.* Convergence of dual objective gradient norms of GSSB, transportation maps for various f-divergences for uniformly random marginals in dimensions $m = 100$ and $n = 100$, uniformly random cost function, and entropic regularization parameter $\epsilon = 100$. The marginals are shown in the subplot (1,1), the cost matrix **C** is shown at (1,2), and the $L^1$ gradient norm of the dual objective (also the $L^1$ marginal error) is shown at (1,3). The second and third rows show the optimal transport maps found by GSA with the corresponding divergence functions.

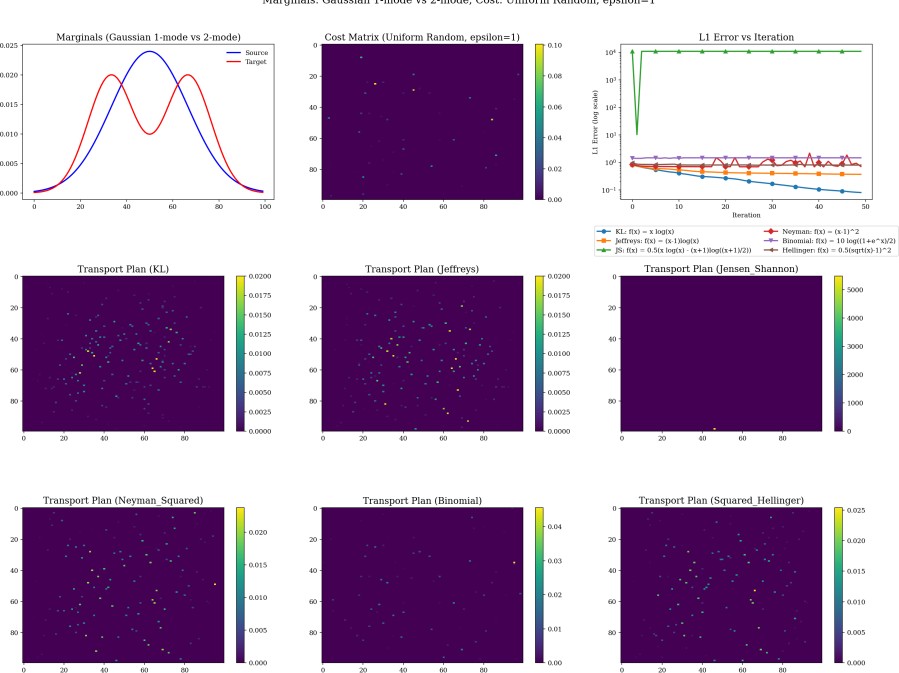

*Figure 13.* Convergence of dual objective gradient norms of GSSB, transportation maps for various f-divergences for Gaussian marginals in dimensions $m = 100$ and $n = 100$, uniformly random cost function, and entropic regularization parameter $\epsilon = 1$. The marginals are shown in the subplot (1,1), the cost matrix $\mathbf{C}$ is shown at (1,2), and the $L^1$ gradient norm of the dual objective is shown at (1,3). The second and third rows show the optimal transport maps found by GSA with the corresponding divergence functions.

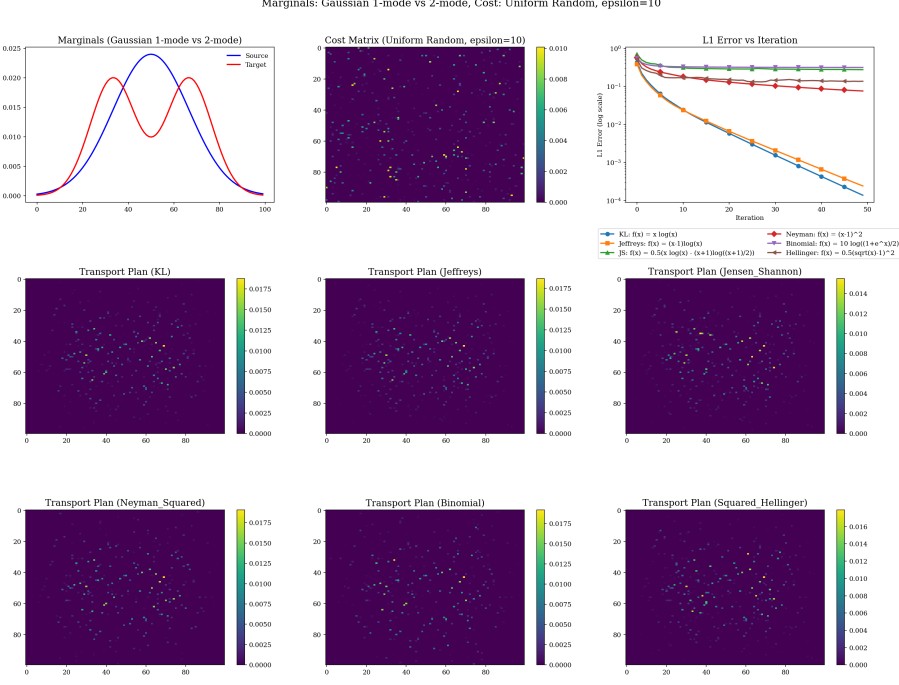

*Figure 14.* Convergence of dual objective gradient norms of GSSB, transportation maps for various f-divergences for Gaussian marginals in dimensions $m = 100$ and $n = 100$, uniformly random cost function, and entropic regularization parameter $\epsilon = 10$. The marginals are shown in the subplot (1,1), the cost matrix $\mathbf{C}$ is shown at (1,2), and the $L^1$ gradient norm of the dual objective (also the $L^1$ marginal error) is shown at (1,3). The second and third rows show the optimal transport maps found by GSA with the corresponding divergence functions.

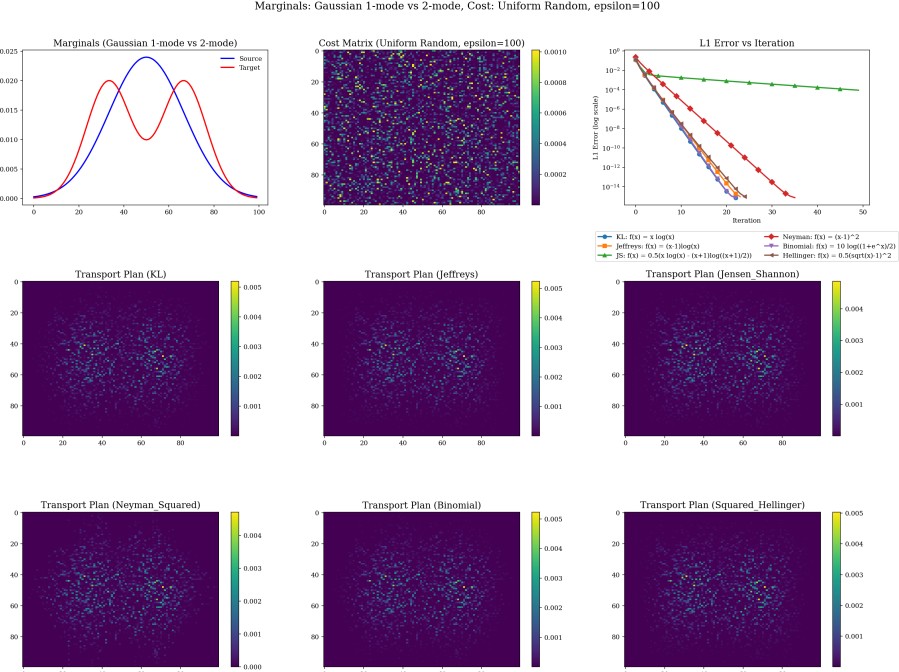

*Figure 15.* Convergence of dual objective gradient norms of GSSB, transportation maps for various f-divergences for Gaussian marginals in dimensions $m = 100$ and $n = 100$, uniformly random cost function, and entropic regularization parameter $\epsilon = 100$. The marginals are shown in the subplot (1,1), the cost matrix **C** is shown at (1,2), and the $L^1$ gradient norm of the dual objective is shown at (1,3). The second and third rows show the optimal transport maps found by GSA with the corresponding divergence functions.

