# OpenReview forum: "Linear convergence of Sinkhorn's algorithm for generalized static Schrödinger bridge"
_ICML.cc/2025/Conference — ICML 2025 poster_

### Official Review · Reviewer_fQdq · 2025-02-17

**Overall Recommendation:** 3

**Summary:**

The authors leverage the process of Lyu and Mukherjee'24 to identify the linear convergence of the generalized Sinkhorn algorithm for a general class of metrics and solve the problem suffered by the KL divergence.

**Claims And Evidence:**

NA

**Essential References Not Discussed:**

NA

**Experimental Designs Or Analyses:**

NA

**Methods And Evaluation Criteria:**

NA

**Other Comments Or Suggestions:**

NA

**Other Strengths And Weaknesses:**

The linear convergence of EOT on bounded cost/ domains is fairly standard. I am unable to determine whether the extension to general metrics is sufficiently significant. I will leave the evaluation of this submission to the other reviewers.

**Questions For Authors:**

Please discuss or clarify the boundedness assumptions.

**Relation To Broader Scientific Literature:**

This paper appears to be a fundamental paper extending the Sinkhorn algorithm to a generalized version supporting more general metrics

**Theoretical Claims:**

The existing linear results assume a bounded domain or bounded cost functions, while yours do not explicitly specify that in the introduction or assumption section? Am I missing something? The authors did assume if f is 2nd differenable on (A, B), I am not sure if this is a standard assumption of boundedness in the optimization community.

I believe this part should be sufficiently clarified or discussed if the boundedness is not required.

---

> ### Author Rebuttal · Authors · 2025-03-28
>
> We thank the reviewer for their positive comments and for raising some great questions.
>
> **Theoretical Claims**
>
> >The existing linear results assume a bounded domain or bounded cost functions, while yours do not explicitly specify that in the introduction or assumption section? Am I missing something? The authors did assume if f is 2nd differenable on (A, B), I am not sure if this is a standard assumption of boundedness in the optimization community. I believe this part should be sufficiently clarified or discussed if the boundedness is not required.
>
> We thank the review for these questions and we will clarify on this point in the revision. Below is our detailed response to the raised questions.
>
> 1. By "domain", we believe the reviewer means the domain of the marginal distributions. Since we consider the discrete generalized static Schrodinger bridge ("GSSB") problem, our domain is always assumed to be bounded.
>
> 2. The EOT setting corresponds to the divergence $f(x)=x\log x$, which is defined on $(0,\infty)=(A,B)$ and is twice differentiable. Our framework of GSSB generalizes the divergence to be arbitrary strictly convex and twice-differentiable function on a domain $(A,B)$, which can be arbitrary. Divergences with bounded domain (e.g., Entropy of Binomial $x$, $f(x)=x\log x + (1-x)\log (1-x)$ on $(0,1)$) has been used for large deviations rate functions of random graphs with given degree sequence. It is less common, however, in the EOT literature. Our GSSB framework unifies both regimes, but their analysis needs separate treatment (see Thm. 2.7 vs. Thm. 2.8).
>
> 3.  The reviewer is correct in that our analysis requires boundedness of cost function, which we expressed in terms of the quantity
> $$
> \kappa := \frac{C_{\max}}{C_{\min}},
> $$
> where $C$ is the cost matrix. Our convergence rate in Thm. 2.6 depends on $\kappa$ and it blows up if $\kappa=\infty$. Specifically in the EOT setting, we have $C_{ij} = \exp(-c(i,j)/\epsilon)$ for a cost function $c$, and $\kappa<\infty$ only if $c$ is bounded. This point was discussed in eq. (23). Furthermore boundedness of the potential functions also depend on the quantity $\kappa$ (see Thm. 2.7, Thm. 2.8).
>
> **Other Strengths and Weaknesses:**
>
> >The linear convergence of EOT on bounded cost/ domains is fairly standard. I am unable to determine whether the extension to general metrics is sufficiently significant. I will leave the evaluation of this submission to the other reviewers.
>
> Thank you very much for raising this fair point. We will clarify this point better in the revision. Below is our detailed response.
>
> 1. Indeed, linear convergence of EOT on bounded cost/domains is well-known in the literature. However, existing convergence analysis and boundedness of potentials along the Sinkhorn iterates depend heavily on the specific structure for the entropic divergence $f(x)=x\log x$ in EOT (e.g., separability of the corresponding dual function $\psi'(\theta)=\exp(\theta)$ leads into close-form Sinkhorn update). Before this work, it remained unclear whether the essential mathematical structure in the theory of Sinkhorn and EOT can be generalize beyond the single particular divergence function. Our work identifies the core mathematical structures that enable general theory and methods.
>
>
>
> 2. As we have discussed in the introduction, there are many important problems in probability, statistics, and combinatorics that can be cast as a generalized static Schrodinger bridge problem. Our work brings the power and the elegance of Sinkhorn algorithm for computing the corresponding potential functions in these problems.
>
>
>
> 3. Another significance is that researchers can now custom-design appropriate divergence function $f$ for their specific domain of application. The entropic divergence $f(x)=x\log x$ comes form the KL-divergence between probability measures, and certainly KL-divergence is one particular choice that can be used. If one tries to find optimal Schrodinger bridges between data such as text, image, or video, researchers are now free to choose the most effective divergence and can even parameterize it and learn it from the training data. In these regards, we believe our contribution also brings some significant practical implications.
>
> 4. We establish linear convergence with sharp sufficient conditions for the rate being dimension-independent, which can be immensely helpful for high-dimensional data such as images and audio.

---

### Official Review · Reviewer_n3cT · 2025-03-08

**Overall Recommendation:** 2

**Summary:**

This paper presents a generalized SSB problem for any divergence and provides a new Sinkhorn-type algorithm to solve this problem. The paper focuses on the formalization of SB in a matrix optimization problem (1). The authors first construct the generalized Kantorovich duality and propose the generalized Sinkhorn algorithm (GSA).

## update after rebuttal

After carefully reading the rebuttal, I still feel that this paper needs full numerical evaluation to verify the authors' claims. Therefore, I would like to keep my current score.

**Claims And Evidence:**

Entropic optimal transport, or SSB, is the standard formulation of regularized OT. This paper deals with a one possible way of generalization where similar results that can be found in EOT literature. In this sense, I think the overall claim of this paper is sound, and overall results can be adopted in the SB theory. However, I have some concern on the significance of the result and justification for general OT researchers regarding necessity to consider GSSB problem (1). The authors are advised to elaborate on the significance of their work for the application side of GSSB.

**Essential References Not Discussed:**

Essential references are well-discussed and properly placed.

**Experimental Designs Or Analyses:**

The experiments for GSA are not provided. Nonetheless, I would like to encourage the authors to at least show basic convergence properties using a (discrete) toy setup for a broader group of audience.

**Methods And Evaluation Criteria:**

This paper does not explicitly propose a method other than the generalized Sinkhorn algorithm. This part must be evaluated in the other section.

**Other Comments Or Suggestions:**

* I would like to suggest the authors show an experiment of GSB (other than entropic regularization) using actual code following analysis Cuturi, 2013.
* There are may possible ways of generalization of OT. For example [1] shows another solid framework for regularized OT problems with mathematical rigor using the Orlicz space. The authors are encouraged to elaborate on contributions of their version of generalization GSA and matrix optimization problem (1).

[1] Lorenz, D., & Mahler, H. (2022). Orlicz space regularization of continuous optimal transport problems. Applied Mathematics & Optimization, 85(2), 14.

**Other Strengths And Weaknesses:**

As I mentioned, I have some questions for the authors of technical details on Theorem 2.6 and 2.8 other than applying some technical SSB results to GSSB in the same manner.

Currently, my review shows some concerns about the significance of GSA since the linear convergence of Sihkorn is well known. I am open to changing my scores when the authors address my concerns in their rebuttal.

**Questions For Authors:**

* If both the Sinkhorn algorithm and GSA show linear convergence (see Carlier, 2022), why do we have to consider GSA, knowing that Sinkhorn can be implemented in a very efficient manner and shows an almost perfect solution with just a few iterations in practice (Cuturi, 2013)?
* It was hard for me to understand recent result of Lyu & Mukherjee, 2024. Could you explain the random matrix theory for Sinkhorn and how you applied the theory into Eq. (5)?

**Relation To Broader Scientific Literature:**

Optimal transport and Schrodinger bridge problems has a lot of applications in other ML problems, and other domains such as scientific modeling, natural language processing, economics.

**Theoretical Claims:**

The main theoretical claim of this paper is in the generalization of linear convergence of Sinkhorn. I think the overall results are interesting and well-presented. However, I would like to stress that the majority of results are from the direct interpretation of well-founded results, especially (Carlier, 2022), (Marino and Gerolin, 2020), and (Lyu & Mukherjee, 2024). Since these results are already known, I would like to point out that the theoretical contributions should be considered by the justification of GSSB and GSA.

---

> ### Author Rebuttal · Authors · 2025-03-28
>
> We thank the reviewer for their insightful and helpful comments.
>
> **Claims and Evidence**
>
> While our work was motivated and inspired by the works the reviewer mentioned, our primary motivation was to identify the core mathematical structures that enable the elegant theory of SA/EOT in a significantly broader setting. In this way we can (1) provide deeper insights on the existing theory of SA/EOT,  (2) bring the powerful and elegant theory of GSA/GSB to many problems in math/stat (that we listed in the introduction), and (3) broaden the already powerful applicability of SA/EOT to GSB problems for new possibility of applications.
>
> For point (1), existing convergence analysis for EOT (e.g.,  Carlier '22) and boundedness of potentials along the Sinkhorn iterates depend heavily on the specific structure for the entropic divergence $f(x)=x\log x$ (e.g., separability of the corresponding dual function $\psi'(\theta)=\exp(\theta)$ leads into close-form Sinkhorn update)  Before this work, it remained unclear whether the essential mathematical structure in the theory of Sinkhorn and EOT can be generalized beyond the single particular divergence function.
>
> For point (2), we discussed wide applicability of our GSB frameworks that include random graphs, contingency tables in statistics, and random matrices in probability in the introduction.
>
> For point (3), advantages of using general $f$-divergence include:
>
> 1.	Asymmetry: KL divergence is asymmetric, leading to different behaviors when swapping distributions. Symmetric f-divergences (e.g., Jensen–Shannon) may be preferable, as seen in many GAN variants.
> 2.	Tail Sensitivity: KL’s logarithmic nature heavily penalizes tail discrepancies, which isn’t always desirable.
> 3.	Robustness: KL is sensitive to support mismatches and outliers, making it less robust in certain applications.
>
> Potential applications of GSSB/GSA include:
>
> 1.	GANs: Many GAN variants use f-divergences (e.g., Jensen–Shannon, Pearson’s ￼) for better stability. Our GSSB extends these to diffusion/generative models like Schrödinger bridge-based approaches.
>
> 2.	Variational Inference: KL divergence in variational inference favors either mode-seeking (forward) or mass-covering (reverse). Using f-divergences (e.g., α or Rényi divergences) allows finer control, especially for multi-modal posteriors, avoiding mode collapse.
>
> **Theoretical Claims**
>
> We thank the reviewer for appreciating our results. We would like to point out some key differences between our work and the references the reviews mentioned.
>
> 1. (Carlier 2022): Please refer to our earlier discussion for point (1).
>
> 2. (Marino and Gerolin 2020) The authors prove asymptotic convergence in the L^p space, but linear convergence was not established. Moreover, again in their work the KL divergence is used, whereas our work addresses for a broad class of divergences.
>
> 3. In (Lyu and Mukherjee 2024) Please see the questions section.
>
>
> **Experimental Designs**
>
> We thank the reviewer for their suggestion. Following this, we add simulations of our GSA method for different f-divergences (https://ibb.co/Q7JRkRF1) with more thorough simulations in the revision.
>
> **Other Strengths and Weaknesses**
>
> We are not entirely sure which details about theorem 2.6 and 2.8 the reviewer refers to.
>
> **Other Comments Or Suggestions**
>
> 1. We are adding experiments for different f-divergences.
>
> 2. We thank the reviewer for bringing the Orlicz space paper [1] to our attention. Our setting is indeed closely related to that in [1], where their Young's function corresponds to our $f$-divergence in the discrete setting. However, their Young's function is required to have at least linear growth, which rules out cases like quadratic $x^2$ (e.g., Gaussian prior) divergence. Our work relaxes the class of divergences by only requiring strict convexity of f. Finally, their T-convergence result is asymptotic, but we establish linear convergence with sharp sufficient condition for the rate being dimension-independent.
>
> **Questions**
>
> 1. Why GSB? Due to the space constraint, we kindly refer the reviewer to our response to reviewer fQdq in "Other Strengths and Weaknesses".
>
> 2. (Lyu and Mukherjee '24) The authors consider margin constrained random matrices with i.i.d. entries from distribution $\mu$ and show that such random matrices are concentrated around a deterministic matrix, called the typical table, in the cut norm. They showed that the typical table is in fact a GSB between the row/column margins with the divergence $f$ being the KL-divergence between the exponential tilting of $\mu$ from $\mu$. When $\mu=Poisson(1)$, the divergence becomes $x\log x - x$, which is equivalent to the EOT divergence $x\log x$ since the total sum is constant over set of matrices with fixed row and column sums. In order to cast the typical table problem into the Sinkhorn setting (5), we only need to compute $(f')^{-1}$. The authors showed that this is simply the log of the Laplace transform of $\mu$.

---

> > ### Comment · Reviewer_n3cT · 2025-04-04
> >
> > I appreciate the authors’ detailed response. Before finalizing my score, I would like to confer with the other reviewers. In the meantime, it would be very helpful if you could share some of the numerical experiments---or, alternatively, details of the experimental design if not possible---via an anonymous link during the rebuttal phase.

---

> > > ### Author Response · Authors · 2025-04-04
> > >
> > > We thank the reviewer for the continued feedback. Regarding the requested numerical experiment,  we wonder if the reviewer might have missed the link to our experimental result that we shared in our previous response.
> > >
> > > “We thank the reviewer for their suggestion. Following this, we add simulations of our GSA method for different f-divergences (https://ibb.co/Q7JRkRF1) with more thorough simulations in the revision.”
> > >
> > > However, we do acknoledge that more extensive experimental results and detailes could have been more helpful. We will share an additional link for this regard as soon as we can.
> > >
> > > **Update on 4/6**
> > >
> > > We thank the reviewer again for the request on further experimental results to supplement our theoretical contribution. While it took longer than we initially expected, we were able to complete extensive experiments on GSA/GSSB, which can be found in the link here: https://drive.google.com/file/d/1_ELa4uGO9riKAX63pqeyfjtx46RlksCA/view?usp=sharing
> > >
> > > We have considered total 12 settings in dimension 100 combining the three hyperparameter settings:
> > >
> > > * Marginal distributions: (1) Each coordinate is uniformly sampled from (0,1) and then normalized; and (2) Gaussian distributions with 1- or 2-modes.
> > >
> > > * Cost functions: (1) Each coordinate is uniformly sampled from (0,1); (2) $L^2$-cost function
> > >
> > > * Entropic regularization: (1) $\epsilon=1$, (2) $\epsilon=10$, and (3) $\epsilon=100$.
> > >
> > > For each setting, we have experimented with the following five divergence functions:
> > >
> > > KL: $f(x) = x\log(x)$
> > >
> > > Pearson: $f(x) = (x-1)^2$
> > >
> > > Jeffreys: $f(x) = (x-1)\log(x)$
> > >
> > > Jensen_Shannon $f(x) = \frac{1}{2}(x\log(x) - (x+1)\log(\frac{x+1}{2}))$
> > >
> > > Binomial: $f(x) = 10\log(\frac{1+e^x}{2})$
> > >
> > >
> > > Each experiment is organized as a (3,3) plots. Marginal distributions are shown in the subplot (1,1), the cost matrix C is shown at (1,2), and the L1 gradient norm of the dual objective (also the L1 marginal error) is shown at (1,3). The second and the third rows show the optimal transport maps found by the GSA with the corresponding divergence functions.
> > >
> > > We observe that using different divergence functions show different linear convergence rates, sometimes the standard KL divergence is not the fastest one. Also, the transport maps show qualitative differences depending the divergence. This indicates the potential benefit of using more adapted divergence functions.
> > >
> > > Please let us know if the reviewer has any further questions and/or comments.

---

### Official Review · Reviewer_U8FC · 2025-03-13

**Overall Recommendation:** 4

**Summary:**

This paper studies the generalized static Schrödinger bridge (SSB) problem, which extends the classical SSB by replacing the entropy divergence with any strictly convex function. Then it derives the Sinkhorn-type algorithm for solving the generalized SSB, and proves the linear convergence of the algorithm. Moreover, this paper examines the conditions under which the convergence rate is independent of the dimensions of the problem.

## Update after rebuttal

After reading the rebuttal, I am satisfied with the further clarifications, and I decide to keep my score.

**Claims And Evidence:**

Yes, the theoretical claims are supported by proofs.

**Essential References Not Discussed:**

None.

**Experimental Designs Or Analyses:**

This is a theoretical paper and does not involve numerical experiments.

**Methods And Evaluation Criteria:**

This is a theoretical paper and does not involve numerical experiments.

**Other Comments Or Suggestions:**

Here are a few additional comments aiming at helping the author(s) further improve the article:
1. Some symbols in Theorem 2.6 may be misused. For example, both $k/k_0$ and $t/t_0$ are used for the iteration number.
2. The same letter is used in different contexts, which may bring confusions to readers. For example, $t$ is the iteration number in Theorem 2.6 but also the bound constant in equation (18). $\varepsilon$ is used in Theorems 2.6 and 2.8 but also the regularization parameter of entropic-regularized optimal transport in equation (22).
3. Typo, above equation (30), $\mu$ should be $\lambda$.
4. This is not mandatory, but it would be interesting to see how the convergence rate of the Sinkhorn algorithm is affected by the regularization parameter $\varepsilon$ in entropic-regularized optimal transport. Such an analysis might be helpful since it is typically believed that a smaller regularization makes the Sinkhorn algorithm converge slower, and a quantitative analysis would provide more insights.

**Other Strengths And Weaknesses:**

I think this paper is a solid contribution to the understanding of Sinkhorn-type algorithms, given that some popular models are special cases of the generalized SSB. The convergence results are also presented in a clear way, which offer many insights on the behavior of algorithms.

**Questions For Authors:**

This is more of a request for clarification: above equation (9), the text says that the concavity of the objective function is not strict. Is it solely due to the one redundant degree of freedom in $(\alpha,\beta)$? Once we remove this redundancy, e.g., by setting $\alpha_1=0$, does the concavity becomes strict?

**Relation To Broader Scientific Literature:**

The analysis provided by this paper may be used to understand the convergence property of other related optimization problems.

**Theoretical Claims:**

I have not checked every detail of the proof, but the overall correctness should be ensured.

---

> ### Author Rebuttal · Authors · 2025-03-28
>
> We thank the reviewer for the positive evaluation of our work and further suggestions.
>
> **Other Comments or Suggestions**
>
> 1. Thank you for pointing out this typo, we shall make the change to refer to iteration count by $k$ instead of $t$.
> 2. In the revision we will maintain $t$ for the upper bound on the tameness condition and use $k$ for the iteration count throughout.
> 3. Noted and changed.
> 4. This is an excellent point raised by the reviewer. In fact, below Thm. 2.8, we have already discussed how our general results specializes to the EOT setting. Carlier '22 shows that the linear convergence rate of Sinkhorn for EOT is proportional to $\exp(\Vert c\rVert_{\infty} /\epsilon)$, where $c$ is the cost function and $\epsilon$ is the entropic regularization parameter. Our Thm. 2.6 (i) shows that Sinkhorn for GSSB has linear convergence rate proportional to the quantity $\kappa$ defined as
> $$
> \kappa := \frac{C_{\max}}{C_{\min}},
> $$
> where $C$ here is the cost matrix. Specifically in the EOT setting, we have $C_{ij} = \exp(-c(i,j)/\epsilon)$, so $\kappa$ becomes
> $$
>  \kappa  = \exp\left( \epsilon^{-1}( \max c(\cdot,\cdot) - \min c(\cdot,\cdot)) \right)\le  \exp(\Vert c \Vert_{\infty}/\epsilon).
> $$
> Thus our convergence rate in Thm. 2.6 shows the same dependence on the regularization parameter $\epsilon$ in the literature, which is inverse exponential. One interesting point is that $\kappa$ above depends only on the span semi-norm of the cost function $c$, which can be significantly smaller than the supermum norm of $c$. But one could WLOG assume $\min c$ is zero, so both quantities are the same.
>
>
> **Questions for Authors**
>
> We thank the reviewer for asking this excellent question. We in fact have tried to establish linear convergence of Sinkhorn for GSSB in this way. In a nutshell, as you correctly pointed out, the dual objective becomes strictly concave once we eliminate one redundant degrees of freedom by, e.g., setting $\alpha_1 = 0$ throughout, so we only have $n+m-1$ dual variables to optimize. Let's assume $m=n$ for simplicity of the discussion. But the main issue in this approach is that the condition number of the Hessian can be as large as order $n$, since the largest eigenvalue of the Hessian is always order $n$ but the smallest eigenvalue can be order 1. In turn, Sinkhorn viewed as alternating maximization for the dual objective restricted in the subspace of $\alpha_1=0$ gives only a dimension-dependent linear convergence rate $(1-O(n^{-1}))^k=\exp(O(k/n))$.
>
> We'd like to provide further details on this point. The Hessian of the dual objective restricted to the subspace $\alpha_1 = 0$ has the following 2 x 2 block structure
> $$
> H(\alpha,\beta) =
> 	\begin{bmatrix}
> 		E_{2\bullet} &   & & E_{21} & \dots & E_{2n} \\\\
> 		& \ddots &  & & \ddots & \\\\
> 		& & E_{n\bullet} & E_{n1} & \dots & E_{nn} \\\\
> 		\hline
> 		E_{21} & \dots & E_{n1} &  E_{\bullet 1} & &  \\\\
> 		& \ddots & &  &\ddots &  \\\\
> 		E_{2n} & \dots & E_{nn} &  & & E_{\bullet n}
> 	\end{bmatrix}
> $$
> where $E_{ij}=\psi''(\alpha_i + \beta_j)$. $E_{i\bullet}$ and $E_{\bullet j}$ denotes the row and column sums, respectively. That the Hessian is positive definite can be seen by Girshgorin's circle theorem, since the diagonal entries are strictly larger than the absolute row sums by a single entry $E_{1j}$. for the first row block, and similarly for the second. The $O(1)$ gap essentially gives that the typical size of the minimum eigenvalue is $O(1)$. On other hand, the maximum eigenvalue is about 2 times the row sums of E, which is of order $n$.
>
> For a concrete example, consider the quadratic divergence $f(x)=x^2/2$, in which case $\psi(\theta)=\theta^2/2$ so $\psi''(\theta)=1$. Hence the Hessian has diagonal entries all $n$ and the off-diagonal blocks have entries all 1. The eigenvalues for this matrix are $n$ (with multiplicity $2n-3$) and $n\pm \sqrt{n(n-1)}$. The same computation holds for general divergence when $\alpha$, $\beta$ are constant vectors.
>
> Thank you very much for your feedback and questions. We are happy to engage in further discussions.

---

> > ### Comment · Reviewer_U8FC · 2025-04-01
> >
> > Thanks for the detailed response, and I want to first ask a quick question regarding the use of Girshgorin's circle theorem. For the upper block of $H(\alpha,\beta)$, the diagonal elements should be exactly equal to the sum of off-diagonal elements. Therefore, it does not exclude the possibility that some eigenvalues may be zero. Is that correct? Of course, the positive definiteness of $H(\alpha,\beta)$ can still be proved, but I feel it is not the direct consequence of the Girshgorin theorem.

---

> > > ### Author Response · Authors · 2025-04-01
> > >
> > > Thanks for the prompt response and follow-up question. It seems like our sketch was oversimplified. Here is a more detailed argument.
> > >
> > > *Claim.* Let $s_*$ denote the minimum eigenvalue of $H$. Then $s^{*}$ is positive and of order $\Omega(1)$.
> > >
> > >
> > > *Proof.*
> > >
> > > The row sums of the (1,2) block coincide with the diagonal entries in the (1,1) block (as you pointed out), but the row sums of the (2,1) block are off from the diagonal entries of the (2,2) block by a single $E_{1j}$ entry. Hence Girshgorin's circle theorem cannot be applied directly, but we can do so with some more work.
> > >
> > >
> > > Let's denote the diagonal blocks of $H$ as $D^1$ and $D^2$, and the (1,2) block as $F$.  Look at the characteristic equation
> > >
> > > $$
> > > 0 = \det(H - t I) = \det(D^1 - tI) \det (D^2 - tI - F^T (D^1 -tI)^{-1} F)).
> > > $$
> > >
> > > Zeros of $\det(D^1 - tI) = 0$ are $E_{i\bullet}$ for $i=2,\dots, m$, which are of order $n$. Hence WLOG we will assume $s_*$ is a zero of the second determinant.
> > >
> > > Denote the matrix in the second determinant at $t=s_*$ as  $K$. Then $K$ is singular so it has zero eigenvalue, so by Gershgorin's circle theorem, we must have $K_{ii}-\sum_{k\ne i} |K_{ik}|\le 0$  for some $i=1,\dots, n$. WLOG assume $E_{\bullet 1}\le \cdots \le E_{\bullet n}$. Write
> > > $$
> > > \left[ F^T  (D^1- t I)^{-1} F  \right]_{i,j}
> > > $$
> > >
> > > $$
> > > =\sum_{k=1}^{m-1} \frac{1}{E_{(k+1) \bullet } - t}  F_{k,i} F_{k,j}.
> > > $$
> > >
> > > Since we are assuming $s_*<  \min\\{ E_{2\bullet},\dots,E_{m\bullet} \\}$,
> > >
> > > $$
> > > 0 \ge 	K_{ii} - \sum_{k=1,\, k\ne i }^{n} |K_{ik}|
> > > $$
> > >
> > > $$
> > > = E_{\bullet i}   - s_* -\sum_{k=1}^{n}  \sum_{\ell=1} ^{m-1} \frac{F_{\ell,i} F_{\ell,k} }{E_{(\ell+1) \bullet} - s_*}
> > > $$
> > >
> > > $$
> > > = E_{\bullet i}   - s_* - \sum_{\ell=1} ^{m-1} \frac{E_{(\ell+1),i} E_{ (\ell+1) \bullet}}{E_{(\ell+1) \bullet } - s_*}.
> > > $$
> > >
> > > The above is equivalent to
> > > $$
> > > \frac{E_{2,i} }{1- \frac{s_*}{E_{2\bullet}}}  + \dots + \frac{E_{m,i} }{1 - \frac{s_*}{E_{m \bullet }}  }\ge E_{\bullet i} - s_*.
> > > $$
> > >
> > > Using $E_{1\bullet} \le E_{2\bullet}\le \dots \le E_{m\bullet}$, we can deduce
> > > $$
> > > 		E_{\bullet i } - E_{1i}	 \ge (E_{\bullet i} - s_*) 		\left( 1 - \frac{s_*}{ E_{1\bullet} } \right).
> > > $$
> > > 	It follows that
> > > $$
> > > s_* ^{2} - s_* \left( E_{1\bullet } + E_{\bullet i}  \right) + E_{1 i} E_{\bullet i} \le 0.
> > > $$
> > >
> > > Solving this quadratic inequality,
> > > $$
> > > s_*  \ge \frac{ E_{1\bullet } + E_{\bullet i}  - \sqrt{ ( E_{1\bullet } + E_{\bullet i} )^{2} - 4E_{1i} E_{\bullet i} } }{2}
> > > $$
> > >
> > > $$
> > > = \frac{ 2E_{1i} E_{\bullet i} }{  E_{1\bullet } + E_{\bullet i} +\sqrt{ ( E_{1\bullet } + E_{\bullet i} )^{2} - 4E_{1i} E_{\bullet i} } }
> > > $$
> > >
> > > $$
> > > \ge \frac{E_{1i } E_{\bullet i} } { E_{1\bullet} + E_{\bullet i} }.
> > > $$
> > >
> > > The RHS above is positive and is of order $\Omega(1)$.  $\square$

---

### Decision · Program_Chairs · 2025-05-01

**Decision:**

Accept (poster)

**Comment:**

The paper introduces a generalized formulation of the static Schrödinger bridge (SSB) problem by replacing the standard entropy divergence with a general strictly convex function. It develops a corresponding Sinkhorn-type algorithm, establishes its linear convergence, and identifies conditions where the convergence rate does not depend on the problem’s dimension. The work is recognized as a solid contribution—especially since many well-known models fit within this generalized framework. Although the mathematical claims are well-supported and aligned with known results in entropic optimal transport (EOT), there are concerns regarding its practical significance for the broader optimal transport community. (Further numerical experiment is suggested) The paper motivates the need for general divergence functions by discussing their advantages over the traditional Kullback–Leibler divergence, including improved symmetry, more balanced tail sensitivity, and enhanced robustness.